# Lower Ordovician synziphosurine reveals early euchelicerate diversity and evolution

Lorenzo Lustri [1] ✉, Pierre Gueriau [1,2] & Allison C. Daley [1] ✉

Euchelicerata is a clade of arthropods comprising horseshoe crabs, scorpions, spiders, mites and ticks, as well as the extinct eurypterids (sea scorpions) and chasmataspidids. The understanding of the ground plans and relationships between these crown-group euchelicerates has benefited from the discovery of numerous fossils. However, little is known regarding the origin and early evolution of the euchelicerate body plan because the relationships between their Cambrian sister taxa and synziphosurines, a group of Silurian to Carboniferous stem euchelicerates with chelicerae and an unfused opisthosoma, remain poorly understood owing to the scarce fossil record of appendages. Here we describe a synziphosurine from the Lower Ordovician (ca. 478 Ma) Fezouata Shale of Morocco. This species possesses five biramous appendages with stenopodous exopods bearing setae in the prosoma and a fully expressed first tergite in the opisthosoma illuminating the ancestral anatomy of the group. Phylogenetic analyses recover this fossil as a member of the stem euchelicerate family Offacolidae, which is characterized by biramous prosomal appendages. Moreover, it also shares anatomical features with the Cambrian euarthropod *Habelia optata*, filling the anatomical gap between euchelicerates and Cambrian stem taxa, while also contributing to our understanding of the evolution of euchelicerate uniramous prosomal appendages and tagmosis.

Euchelicerata is a vast clade of mostly predatory arthropods comprising extant forms such as arachnids (the group including scorpions, spiders, mites and ticks) and their closest relatives, the xiphosurans (horseshoe crabs), as well as the extinct eurypterids (sea scorpions), chasmataspidids and the synziphosurines. While phylogenomic studies[1] and fossil evidence[2,3] assisted our understanding of the relationships between the crown-group euchelicerates, the origin and early evolution of euchelicerates remain poorly documented. Euchelicerates have traditionally been united by the presence of specialized frontal-most appendages, the so-called chelicerae, and by the tagmatization of the body into a prosoma, grouping the sensory organs and the walking limbs, and an opisthosoma, bearing book gill opercula[4,5], but some researchers emphasize this last anatomical feature of book

gill opercula may be the only diagnostic characteristic defining Euchelicerata[6], a hypothesis that has been debated[7–9]. There is also no consensus on either the sister group to Euchelicerata or their relationships with Cambrian stem euarthropods. Several Cambrian sister taxa have been proposed for euchelicerates. Particularly: (i) Megacheira, with their subchelate "great appendages" suggested to be homologous structures from which chelicerae were derived[10]; (ii) Vicissicaudata, based on phylogenetic analyses[11]; and (iii) Habeliida[12], owing to the head tagmosis and pseudotagmosis of *Habelia optata*[12]. Moreover, the anatomy of the synziphosurines[13], a Silurian to Carboniferous paraphyletic[14] or polyphyletic[4] group of euchelicerates, is often only incompletely known, particularly with regard to their ventral anatomy, strongly limiting comparisons with the proposed

[1]Institute of Earth Sciences, University of Lausanne, Géopolis, Lausanne, Switzerland. [2]Université Paris-Saclay, CNRS, ministère de la Culture, UVSQ, MNHN, Institut photonique d'analyse non-destructive européen des matériaux anciens, Saint-Aubin, France. ✉e-mail: lorenzo.lustri90@gmail.com; allison.daley@unil.ch

possible sister taxa of stem chelicerates. Two notable exceptions are the basal-most taxa *Offacolus kingi*[15,16] and *Dibasterium durgae*[17] from the Silurian of Herefordshire. These two species possess elongate chelicerae and a peculiar limb arrangement in the prosoma with biramous appendages comprising stenopodous exopods. Since their description, they have been consistently retrieved as basal euchelicerates in phylogenetic analyses[3,4,6,12,18,19].

Here we describe *Setapedites abundantis* gen. et sp. nov., a minute synziphosurine from the Lower Ordovician (late Tremadocian, ~478 million years ago) Fezouata Shale of Morocco[20,21]. The Fezouata Shale is a major marine fossil site with exceptional preservation of labile parts that is transitional between the Cambrian Explosion and the Great Ordovician Biodiversification Event (GOBE) and has provided critical insights into our understanding of the early evolution of metazoans and the structure of the Early Paleozoic marine biosphere[21-25]. *Setapedites abundantis* is one of the most abundant components of this unique assemblage, with hundreds of specimens housed in two main institutions: the Musée cantonal de géologie Lausanne, Switzerland, (MGL) and the Yale Peabody Museum, New Haven, CT, USA (YPM). Phylogenetic analyses recover *Setapedites abundantis* in a clade of stem euchelicerates together with *Offacolus kingi* and *Dibasterium durgae*. It shares several homologies with the Cambrian euarthropod *Habelia optata*, including bipartite opisthosomal tergites, proximal anatomy of the opisthosomal appendages, and possibly an anal pouch. *Setapedites abundantis* documents how euchelicerate uniramous prosomal appendages were derived from the appendages of a habeliid ancestor[12] and illuminates the evolution of early tagmosis in euchelicerates.

## Results

### Systematic Paleontology

Euarthropoda Lankester, 1904[26]
Arachnomorpha Størmer, 1944[27]
Euchelicerata Weygoldt & Paulus, 1979[5]
Offacolidae Sutton, Briggs, Siveter, Siveter & Orr, 2002[16]

**Emended diagnosis.** Euchelicerates with elongate chelicerae, at least second to fifth prosomal appendage pairs being biramous and with reduced gnathobasipods and composed of stenopodous exopods of six podomeres bearing a brush-like group of long and radially arranged setae on the distalmost podomere. Seventh pair of appendages (sixth post-cheliceral appendage pair) uniramous and lobate paddle-like, fringed by lateral spines. Tip of the telson bifurcate. Diagnosis modified from Sutton, et al[16].

*Included taxa. Dibasterium durgae*[17], *Offacolus kingi*[15], *Setapedites abundantis*

*Setapedites abundantis* gen. et sp. nov. (Figs. 1–5)

**Etymology.** The genus name derives from the Latin "seta" and "pedes" referring to its brush-like group of setae on the distalmost podomere of the prosomal stenopodus exopods. The species name derives from the Latin word for abundant "abundans" referring to the abundancy of the species in the Fezouata Shale.

**Diagnosis.** Offacolid with prosomal appendages 2 to 6 homonomous. Fully expressed first opisthosomal segment, and a developed axis from the second tergite to the sixth, bearing sub-axial nodes. Axis of the pre-abdominal tergites transversely bipartite in a less sclerotized anterior part and a more sclerotized posterior part. 11th segment bearing an ovoid ventral process, and ending posteriorly in two pairs of spines flanking a long needle-like telson with a bifurcate tip.

**Material.** Holotype MGL.107741 (Fig. 3A–D, see Supplementary Fig. 2A for normal light photo); paratypes MGL.102899 (part; Figs. 1A and 1B, see Supplementary Fig. 4A for counterpart and Supplementary Fig. 4C and 4D for anatomical details), MGL.102828 (part; Figs. 1C and 1D), MGL.102872 (part; Figs. 1E and 1F), YPM IP 517932c (counterpart; Fig. 2A–D, see Supplementary Fig. 3E and F for anatomical details) YPM IP 517932 (part; Supplementary Fig. 2B and 2C), MGL.102934 (part; Figs. 2E and 2F, see Supplementary Fig. 2G for full specimen and Supplementary Fig. 2I for detail of biramous appendages), MGL.102634 (counterpart; Figs. 2G and 2H, see Supplementary Fig. 2E for full counterpart specimen and 2 F for full part specimen, see Supplementary Fig. 3G and 3H for anatomical details), MGL.102800a (part; Figs. 2I and 2J, see Supplementary Fig. 2H for full specimen and Supplementary Fig. 3A and 3B for anatomical details), MGL.102637 (part; Fig. 4A–E), MGL.102952 (part; Supplementary Fig. 1A and 1B), MGL.102800b (part; Supplementary Fig. 1C and 1D), MGL.102902 (part; Supplementary Fig. 2E and 2F), MGL.102469 (part; Supplementary Fig. 2G and 2H), MGL.102247a (part; Supplementary Fig. 2D, Supplementary Fig. 3C and 3D), MGL.102690 (part; Supplementary Fig. 4B and 4E), MGL.102841 (part; Fig. 4F–K).

**Locality and age.** Fezouata Shale, Zagora province, Morocco (detailed locality information is curated with the specimens). Early Lower Ordovician, late Tremadocian, *Araneograptus murrayi* bio-zonation.

**Remarks.** *Setapedites abundantis* gen. et sp. nov. (Figs. 1, 2, 3, 4 and 5) is superficially similar to *Offacolus kingi* Orr et al.[15] and *Dibasterium durgae* Briggs et al.[17] from the Silurian Herefordshire Konservat-Lagerstätte[16,17], yet it differs from both by its fully expressed first opisthosomal tergite and its bipartite tergal axis. Furthermore, the prosomal shield anatomy of *Setapedites abundantis* strictly resembles that of *Offacolus kingi* (with the presence of a sunken region and a median ridge), while *Dibasterium durgae* is devoid of any dorsally expressed characters on the prosoma. The opisthosoma of *Setapedites abundantis*, on the other hand, more closely resembles that of *Dibasterium durgae*, with the same number of tergites and the same overall dorsal anatomy (see Supplementary information for discussion about anatomical comparison with synziphosurines other than offacolids). Among Cambrian arthropods, the dorsal morphology of *Jianshania furcatus* Luo et al[28]. from the Chengjiang biota appears most similar to that of *Setapedites abundantis*[28,29] (see Supplementary information for additional discussion about a possible relationship linking *Jianshania furcatus* with *Setapedites abundantis* and its implications).

### Description of the Fezouata Shale synziphosurine

*Setapedites abundantis* possesses an elongate, dorsoventrally flattened body, divided into an anterior prosoma bearing a fused dorsal headshield, and an unfused opisthosoma clearly differentiated into (medially) a pre-abdomen and (posteriorly) an abdomen (Figs. 1A and 1B). Its total length varies between 4.33 and 6.5 mm (excluding appendages and telson), its maximum width (prosoma) between 2.23 and 2.9 mm (see Supplementary Table 1 for list of measurements).

The headshield is semi-circular in outline and wider than long (Figs. 1C and 1D). It projects anteromedially into a minute prosomal spine (ps in Supplementary Fig. 1C and 1D) and latero-posteriorly into a pair of small, ventrally directed genal spines (gs in Supplementary Fig. 1C and 1D) often partially overlapping the tergopleura of the first opisthosomal segment in dorsal view. Specimens preserved in lateral view show that the headshield is not flat, but domed (Figs. 1E and 1F, Supplementary Fig. 1A and 1B). The prosoma bears a narrow semi-circular rim along the anterior margin preserved as a thickened cuticular structure (pr in Fig. 1E and 1F) and a median ridge extending from the posterior to the anterior prosomal margins (mr in Figs. 1E and 1F). The median ridge is flanked on each side by slightly concave regions (sunken region) (sr in Fig. 1E and 1F, Supplementary Fig. 1A and 1B). The headshield ventrally folds into a doublure that widens antero-medially (db in Fig. 2B, Supplementary Fig. 1E and 1F). On the ventral side, a centrally-positioned structure composed of a small, rounded plate

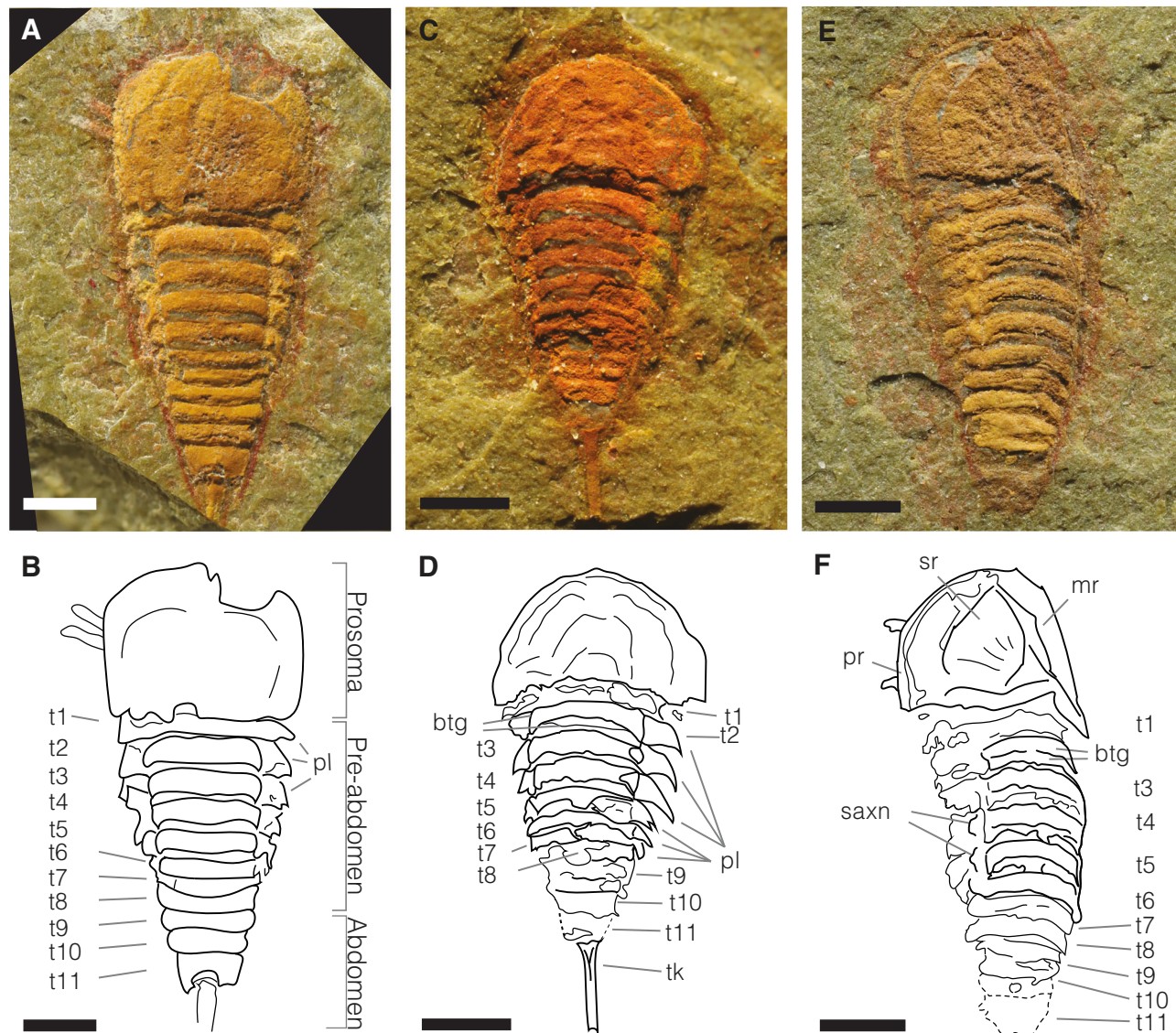

**Fig. 1 | Dorsal anatomy of *Setapedites abundantis* gen. et sp. nov.**
**A**, **B** MGL.102899 and interpretative drawing, articulated specimen in dorsal view.
**C**, **D** MGL.102828 and interpretative drawing, articulated specimen in dorsal view.
**E**, **F** MGL. 102872 and interpretative drawing, articulated specimen in dorsal view.

Abbreviations: btg, bipartite tergites; mr, median ridge; pl, pleura; pr, prosomal rim; saxn, sub-axial node; sr, sunken region; t1–11, tergites 1–11; t, telson; tk, telson keel. Scale bars, (**A**–**F**) 1 mm.

aligned antero-posteriorly can be identified as a labrum (lb in Figs. 2A–D, 2I and 2J, Supplementary Fig. 2B and 2C, see Supplementary information for additional discussion about the labrum in *Setapedites abundantis*), presumably covering the mouth. No eyes or ophthalmic region are visible.

The prosoma bears six pairs of appendages, which insert around the labrum (Figs. 2A and 2B, 2E–2J, Figs. 3A and 3B, Fig.4). The first pair of appendages are uniramous, elongate chelicerae, anteroventrally oriented (gray highlighted appendages in interpretative drawings; Fig. 2A–D, Fig. 2I–2J, Figs. 3A and 3B, Fig.4, Supplementary Fig. 1E–1H, Supplementary Fig. 2B and 2C). Evidence for a chelate last podomere in the chelicerae is provided by a single specimen (gray highlighted appendages in interpretative drawings; Fig. 2A–D, Supplementary Fig. 2B and 2C), otherwise they are preserved retracted and bent under themselves (Figs. 2I and 2J, Figs. 3A and 3B). Appendages 2 to 6 are biramous and lateroventrally oriented (Fig. 2E–K, Figures 3A and 3B, Fig. 4). The endopods are often preserved retracted and bent back onto themselves (blue appendages in Figs. 2I and 2J, Figs. 3A and 3B and Fig. 4); nevertheless, their detailed anatomy is preserved. At least

one specimen shows evidence for a chelate last podomere in the post-cheliceral endopods (che in Figs. 2I and 2J, Supplementary Fig. 3A and 3B). No count of the podomeres is possible for the endo-pods. The exopods are divided into six podomeres (1–6 in Figs. 2G and 2H, Supplementary Fig. 3G and 3H). The sixth podomere bears brush-like setae (bst in Figs. 2G and 2H, Supplementary Fig. 2D-F, Supple-mentary Fig. 3C and 3D and Supplementary Fig. 3G and 3H). The fifth podomere bears a couple of setae ventrally (st in Figs. 2G and 2H, Supplementary Fig. 3G and 3H) and the fourth one bears one single seta on the opposite side (ss in Figs. 2G and 2H, Supplementary Fig. 3G and 3H). The rami of the biramous appendage attach to a protopodite (basipodite) of rectangular shape (bs in Figs. 2E and 2F, Supplementary Fig. 2G and 2I).

The opisthosoma, 1.5 times longer than the prosoma, tapers posteriorly and consists of eleven somites (each opisthosomal somite bearing a tergite; t1–t11 in Figs. 1B, 1D and 1F and Supplementary Fig. 1G and 1H dorsally), is divided into a pre-abdomen comprised of the first eight somites and bearing pleural spines (pl in Fig. 1A-D), and an abdomen made up of the last three somites plus a needle-like telson

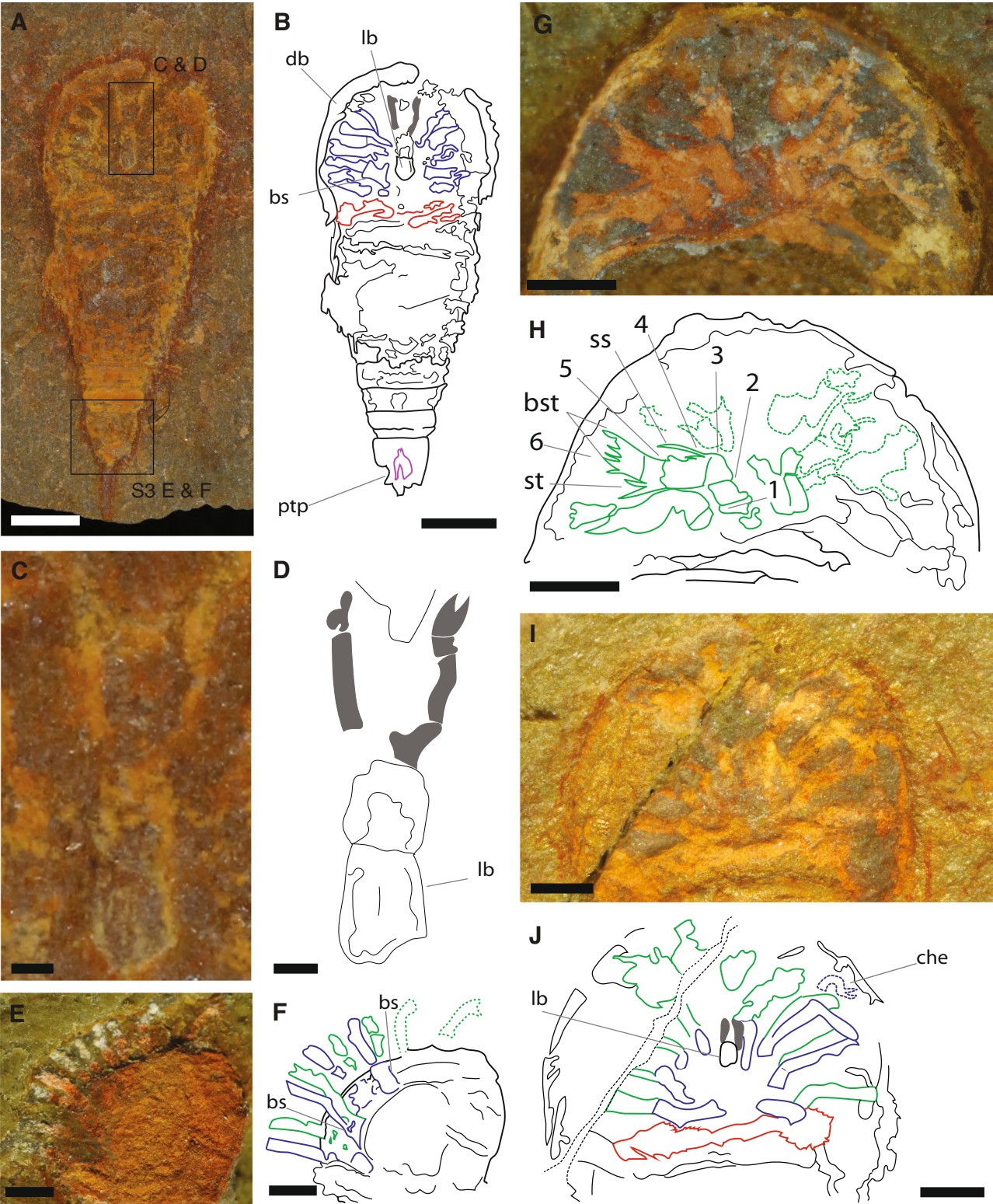

**Fig. 2 | Prosomal appendicular anatomy of *Setapedites abundantis* gen. et sp. nov. A**, **B** YPM IP 517932c and interpretative drawing (counterpart), articulated specimen in ventral view. **C**, **D** YPM IP 517932c and interpretative drawing, chelicerae, and labrum anatomy detail. **E**, **F** Close-up of the prosoma of MGL.102934 and interpretative drawing, in dorso-lateral view. **G**, **H** Close-up of the prosoma of MGL.102634 and interpretative drawing, in ventral view. **I**, **J** Close-up of the prosoma of MGL.102800a under alcohol and polarized lighting, and interpretative drawing, in ventral view. Abbreviations: 1–6, podomeres 1–6 of the exopod; ptp, pretelsonic process; bs, basipodite; bst, brush-like setae; che, chelate podomere; db, doublure; lb, labrum; ss, single setae; st, pair of setae. Chelicerae are highlighted in gray, endopods in blue, exopods in green, opisthosomal appendages in red, and the pretelsonic process in purple. Scale bars, (**A**, **B**) 1 mm; (**C**, **D**) 100 μm; (**E–K**) 500 μm. *See also* Supplementary Fig. 2B, 2E–H *for full views of specimens MGL.102934, MGL.102634 and MGL.102800a and YPM IP 517932c part.*

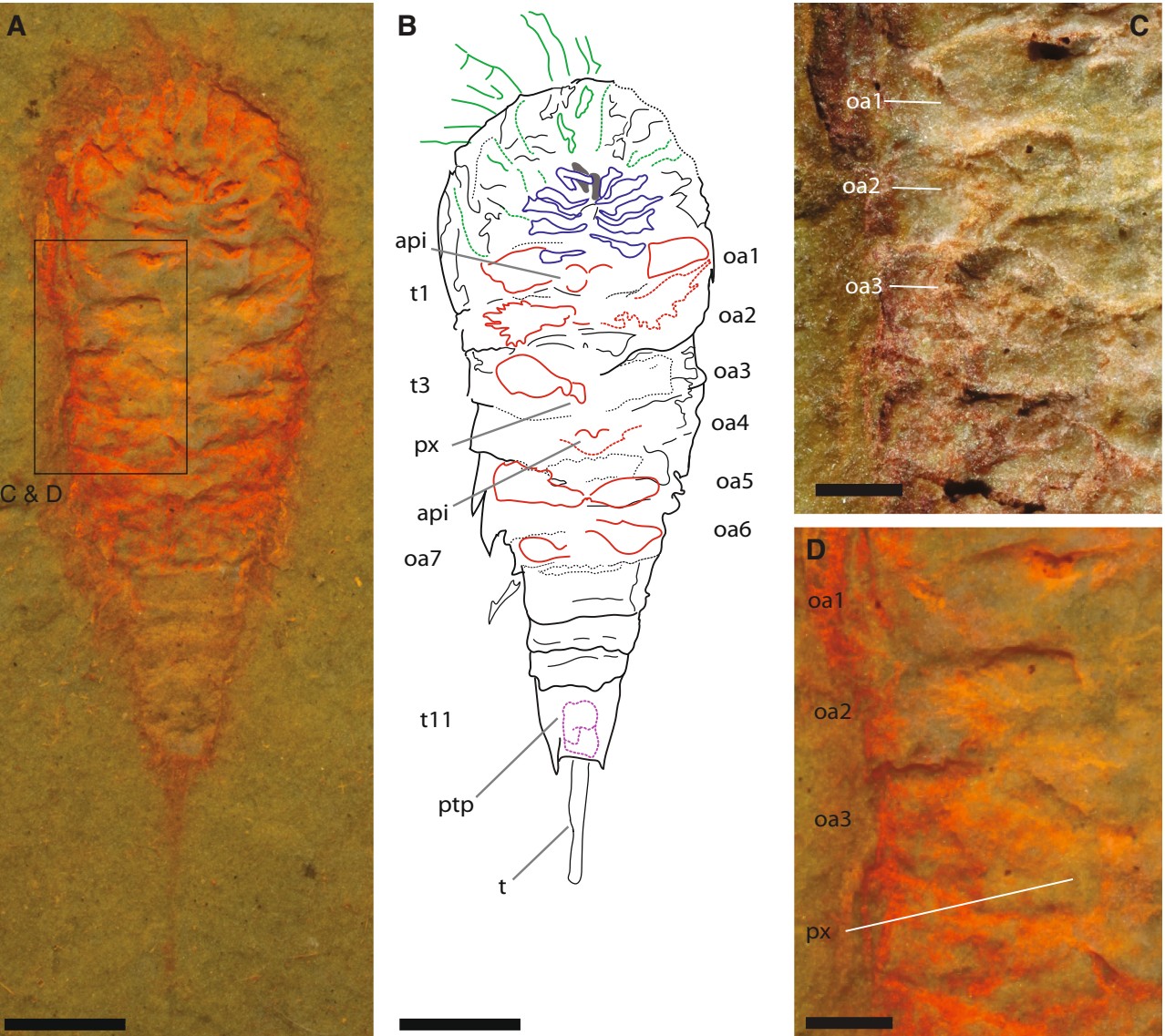

**Fig. 3 | Opisthosomal appendicular anatomy of *Setapedites abundantis* gen. et sp. nov. A, B** MGL.107741 and interpretative drawing, holotype, articulated specimen in ventral view photographed under alcohol and with polarized filter. Composite line drawing from E and Supplementary Fig. 2A. (**C, D**) MGL.107741, Figs. 2E and 2F, opisthosomal appendages details, **C** with polarized filter and **D** under alcohol and polarized filter. Abbreviations: api, appendages insertion; oa1–7, opisthosomal appendages 1–7; ptp, pretelsonic process; px, proximal part of opisthosomal appendage; t1–11, tergites 1–11; t, telson. Chelicerae are highlighted in gray, endopods in blue, exopods in green, opisthosomal appendages in red, and the pretelsonic process in purple. Scale bars, (**A, B**) 1 mm; (**C, D**) 200 μm. *See also* Supplementary Figs. 2A *for full views of specimens MGL.1027741 polarized light photo.*

(of which only the last somite preserves modified pleural spines) (Figs. 1A and 1B, Supplementary Fig. 1A–D and Supplementary Fig. 4). The first tergite (tergite 1) is fully developed but partially overlapped by the prosomal shield and does not present a clear distinction between tergopleura and axis (Fig. 1A–F). In some specimens tergite 2 is slightly hypertrophic (Figs. 1A and 1B). Tergites 2–6 have an inflated axis, which constitutes the most prominent part of the body (Fig. 1A–F). The axis has squared lateral margins and antero-posterior bipartite sclerotization, the anterior part being less sclerotized (btg in Fig. 1C–F). A transverse margin divides the anterior and posterior portions of this structure. However, the anterior portion is often less well-preserved than the posterior portion due to its lower degree of sclerotization. Tergites 2–8 bear a couple of sub-axial nodes on the lateral edges of the axis, which connect the axis with the tergopleura (saxn in Figs. 1E and 1F). The width of the tergites decreases slightly from tergites 3 to 8. Tergites 1 to 8 bear leaf-shaped tergopleura,

starting with a wide attachment to the axis and ending in a pointed spine directed ventro-proximally (pl in Fig. 1A–D, Supplementary Fig. 1C and 1D).

In the pre-abdomen, somites VII to XIII bear a pair of appendages, while somite XIV is devoid of appendages (oa1-7 in Fig. 3 and Supplementary Fig. 1G and 1H). Opisthosomal appendages possess a distinct proximal part (px in Figs. 3A and 3B). Appendages of pre-abdominal somite VII are uniramous and paddle-like (Fig. 3 and Supplementary Fig. 1G and 1H) and insert medially (api in Figs. 3A and 3B). Appendages 2–6 are rarely preserved but are uniramous and have similar insertions as appendage 1 (Fig. 3 and Supplementary Fig. 1G and 1H). The abdomen, which is half the length of the pre-abdomen, is composed of three somites (t9–11 in Figs. 1A and 1B) without appendages. The first two abdominal somites lack pleural spines. The pretelson is twice as long as the other opisthosomal somites and carries posteriorly-directed tergopleura modified into two pairs of spines flanking the telson

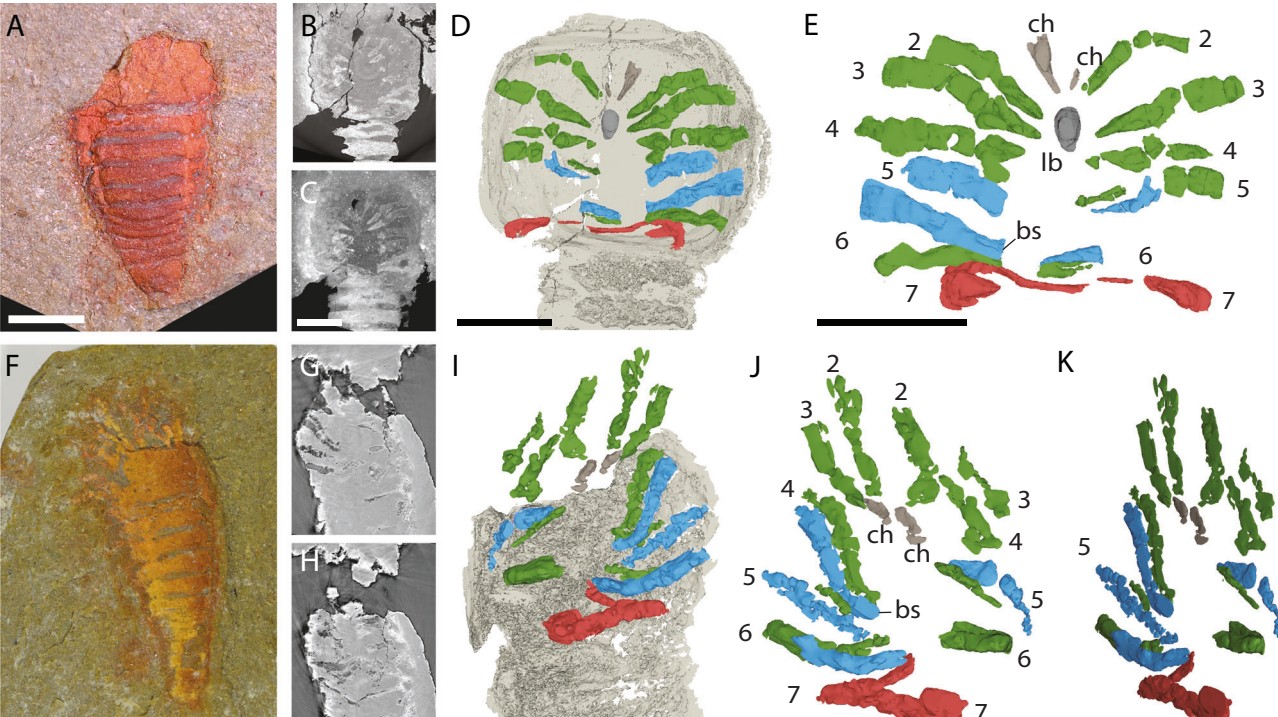

**Fig. 4 | Prosomal appendages of *Setapedites abundantis* gen. et sp. nov. (MGL.102637a and MGL.102841) based on synchrotron X-ray computed microtomography. A** optical photograph of MGL.102637a, in dorsal view. **B**, **C** single tomogram (**B**) and max intensity projection of 52 tomograms (**C**) through the prosomal appendages. (**D**, **E**) 3D segmentation rendering of the prosomal appendages in ventral (**D**) and dorsal (**E**) views. **F** optical photograph of MGL.102841, in dorso-lateral view. **G**, **H** two single tomograms through the prosomal appendages. **I**–**K** 3D segmentation rendering of the prosomal appendages in ventral (**I**), dorsal (**J**) and dorso-lateral (**K**) views. Abbreviations: 2–7, appendages 2–7; bs, basipodite; ch, chelicerae; lb, labrum. Chelicerae are highlighted in gray, endopods in blue, exopods in green, opisthosomal appendages in red. Scale bars 1 mm (scale bars in **C** and **H** apply for **B** and **G**, respectively).

(ls in Supplementary Fig. 1C and 1D). Each abdominal somite is encased by a fused tergite and a sternite forming a ring-shaped structure (Fig. 1A and 1B, Supplementary Fig. 1). A small symmetrical rounded process (pretelsonic process) is often preserved under tergite 11, in some cases appearing longitudinally bisected and ovoid in shape (ptp in Figs. 2A and 2B, Figs. 3A and 3B, Supplementary Fig. 1A–F, Supplementary Fig. 3E and 3F) which could represent an anal pouch.

The terminal telson, as long as the pre-abdomen, is narrow and straight, and ends in two spines directed posteriorly (bt in Supplementary Fig. 4). It is slightly triangular in section, and some specimens show evidence for the presence of a median keel (tk in Figs. 1C and 1D). It is internally articulated with the last abdomen segment by an enlarged head (Fig. 1A and 1B).

### The Fezouata Shale synziphsurine is a stem euchelicerate

Four different phylogenetic analyses were conducted to determine the affinity of *Setapedites abundantis*. We first coded it into the matrix from Aria & Caron[6], to assess its broad position among euarthropods. Bayesian phylogenetic analyses were performed and *Setapedites abundantis* is found alongside *Offacolus kingi* and *Dibasterium durgae*, as part of the family Offacolidae, in a sister group position relative to all other euchelicerates (Supplementary Fig. 5, 9 and 10: see "Methods" and Supplementary Information for full phylogenetic methods). We then performed a second analysis focusing on Panchelicerata ( = Chelicerata+Habeliida), using the matrix from Lamsdell[4] that includes numerous synziphosurines and euchelicerates, and to which we added *Habelia* and *Mollisonia plenovenatrix*. Parsimony (implied and equal weighting) and Bayesian phylogenetic analyses were both performed, and *Setapedites abundantis* is found again in a clade with *Offacolus kingi* and *Dibasterium durgae*, inside Euchelicerata and as a sister group to Prosomapoda (*sensu* Lamsdell[4]) (Fig. 6 and Supplementary Fig. 6–8, 11–15: see "Methods" and Supplementary Information for full phylogenetic methods). In the first analysis, Offacolidae is united in possessing the following synapomorphies: short and stout stenopodous prosomal exopods ending in a setal brush, and post-antennular appendages chelate or sub-chelate. The synapomorphies supporting Offacolidae in the second set of analyses are: sixth post-antennular appendage flap-like, seventh to thirteenth post-antennular appendages lacking the endopods, and bifurcated telson tip.

## Discussion
### Stem and sister group chelicerate relationships
The fossil record has provided multiple options for the sister group of total-group Chelicerata (Pycnogonida+Euchelicerata), including megacheirans[30–32] and artiopods[11,33]. In the phylogenetic analyses performed by Legg et al.[11], Vicissicaudata resolved as the closest group to Chelicerata among the artiopods. Among Cambrian euarthropods, *Mollisonia plenovenatrix* Walcott, 1912[34] has been described as possessing a pair of short chelicerae and proto-book gills composed of overlapping exopod flaps, and retrieved as a basal chelicerate[6]. Counter arguments put forward in the literature against *Mollisonia plenovenatrix* being a basal chelicerate cite the poor preservation of these features and functional considerations (chelicerae too small and far from the mouth)[9], the organization of the central nervous system in *Mollisonia symmetrica*[7] (but see[35]), as well as the origins and development of the last three segments complex (pygidium) in the genus *Thelxiope* and *Mollisonia*[8]. To reflect the debate on this taxon and its relationship with Chelicerata, we coded matrices with different character codings for controversial anatomical structures, and there was no impact on the conclusions of the present work regarding the

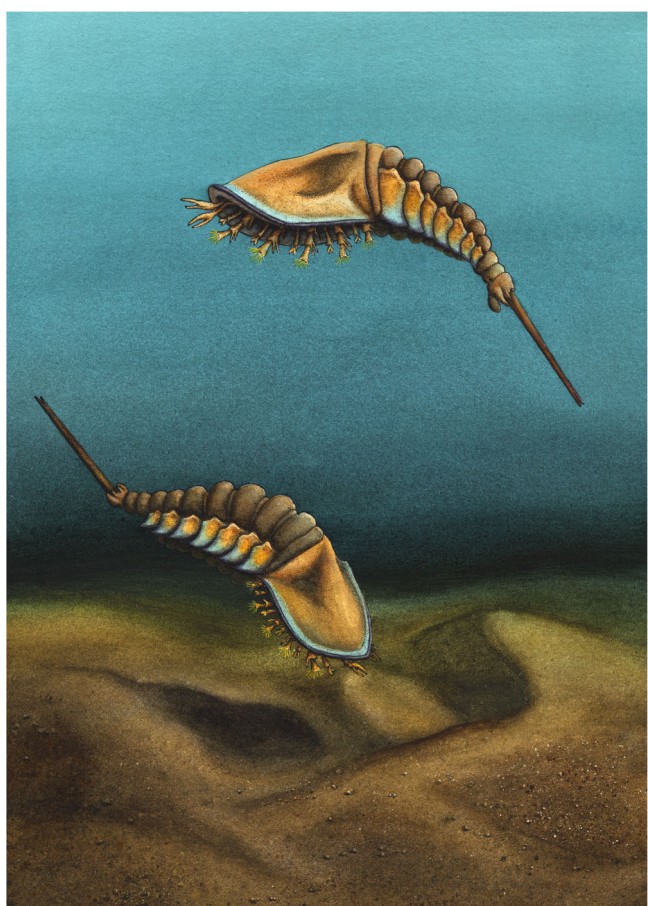

**Fig. 5 | Life reconstruction of *Setapedites abundantis* gen. et sp. nov.** Illustration by Elissa Sorojsrisom.

phylogenetic position of *Setapedites* and the establishment of Offacolidae (for comparisons see Supplementary Fig. 5 and 10 and Supplementary Fig. 6 and 11). Within crown-Chelicerata, Pycnogonida has been retrieved as the sister group of Euchelicerata in phylogenetic analyses based on both molecular and anatomical characters[11,36,37]. Such a Chelicerata clade is supported mainly by the presence of a pair of chelifores in Pycnogonida, considered homologous with the chelicerae of Euchelicerata. Other analyses recovered the pycnogonids nested among the Euchelicerata[38,39]. Aspects of the highly derived anatomy of pycnogonids, such as their uniramous appendages and four segments in the head tagma, cast doubt on this sister-group relationship (Pycnogonida+Euchelicerata)[39,40].

These doubts have been further increased by the emergence of Habeliida as a potential alternative to Pycnogonida as the closest sister group for Euchelicerata[12,41]. *Setapedites abundantis* shows further anatomical similarities that link *Habelia optata* with *Offacolus kingi* and *Dibasterium durgae*, supporting the placement of Habeliida as sister group to Euchelicerata[12]. Resolving the phylogenetic position of pycnogonids awaits new investigations and fossil discoveries to clarify their position with respect to *Habelia optata* and offacolid euchelicerates. Even in the absence of a firm phylogenetic position for pycnogonids, *Setapedites abundantis* contributes to our understandings on the origin and early evolution of two key euchelicerate characters: the transition from biramous to uniramous prosomal appendages, and body tagmosis.

### Nature of the prosomal appendages in euchelicerates

The stenopodous prosomal exopods and the uniramous first pair of appendages in *Habelia optata* are homologies shared with Offacolidae,

linking *Habelia optata* to the Euchelicerata[12]. *Setapedites abundantis* shares other similarities with *Habelia optata*. Besides the stenopodous prosomal exopods, which already led to the hypothesis that *Dibasterium durgae*, *Offacolus kingi,* and *Habelia optata* were closely related[12], and two spines on their preterminal podomeres, *Setapedites abundantis* and *Habelia optata* are further united by their bipartite opisthosomal tergites, and a possibly homologous pretelsonic process (see Supplementary Information for additional discussion about the pretelsonic process). Biramous prosomal appendages with specialized exopods (with respect to the ancestral condition for stem euarthropods[42]), together with a uniramous first pair of appendages (chelicerae) and uniramous appendages of somite VII made of bipartite paddle-like exopods, can therefore be considered as part of the euchelicerate ground plan.

The first pair of uniramous appendages in *Setapedites abundantis*, although rarely preserved, appear to be composed of few elongated articles, confirming their elongated chelicera interpretation in other offacolids and discharging the antenna-like hypothesis[41] (Fig. 2C and 2D). The morphology of prosomal biramous appendages 2 to 5 is consistent within Offacolidae, too, with brush-like exopods nearly identical between *Setapedites abundantis*, *Dibasterium durgae* and *Offacolus kingi*[15–17]. This is especially true for *Setapedites abundantis* and *Offacolus kingi* exopods 2 to 5 considering the exopods shown in Fig. 2G–H. The mesial position of the exopods could cast some doubts on their exopodial nature, however both endopods and exopods have their proximal-most parts very close to each other (visible in Fig. 4D–J) and this has been supposed to be the case in other offacolids too[15–17]. The length of the most well-preserved exopod in this specimen (see also Supplementary Fig. 3) also appears shorter than the exopods preserved in other specimens and extends outside the prosomal shield (e.g., Fig. 1A and 1B), which is likely owing to its preservation in a bent position with an angel parallel to the matrix and the probability of it being the fourth or even the fifth exopod. Prosomal appendages 6 differ in *Offacolus kingi*, with tendril-like exopods, and in *Dibasterium durgae*, with uniramous pushing-legs[15–17]. Exopodial setal brushes are also present in *Habelia optata*[12]. However, they are short and confined to the inner margin of the exopod in *Habelia optata,* whereas they are long and radially arranged from the inner to the outer margins in Offacolidae, clearly differentiating these features. *Mollisonia plenovenatrix* also shows a single exopod in one specimen with a termination made of setae[6], but their precise arrangement is unknown. Possessing exopodial setal brushes may be homologous amongst these taxa, but we interpret the precise morphology of stenopodous exopods 2–5 ending in a brush-like group of long and radially arranged setae to be an autapomorphy of Offacolidae, a conclusion which is supported by our phylogenetic analyses (Fig. 6, Supplementary Figs. 5–9). Similarly, the homonomous appendage series 2–6 found in *Setapedites abundantis* is here considered to be the ancestral state for Offacolidae, and uniramous appendages 6 of *Dibasterium durgae* and *Offacolus kingi* a derived state for Offacolidae. Even if this statement is not supported by our Bayesian and implied weight phylogenetic analyses (Fig. 6, Supplementary Figs. 5, 6, 8 and 9) this can be explained by a loss of the external rami in *Offacolus kingi* and *Dibasterium durgae* independently. An alternative explanation is a non-optimal resolution of the ingroup Offacolidae, as suggested by our equal weight analyses, resulting in a polytomy of the inner relationships of the clade Offacolidae (Supplementary Fig. 7). Briggs et al.[17] suggested the involvement of the gene Distal-less in the expression of the external rami of *Offacolus kingi* and *Dibasterium durgae* and interpreted their limb without a common base as an early evolutionary step towards the complete loss of the upper rami in crown-group euchelicerates from a plesiomorphic condition with limbs sharing a common base[6,12,17,43]. A potential exception is the flabellum of xiphosurids, if further analyses provide support for an exopodial origin rather than an epipodial origin[42,44]. The evidence for the attachment of prosomal biramous limbs onto a common basipodite in *Setapedites*

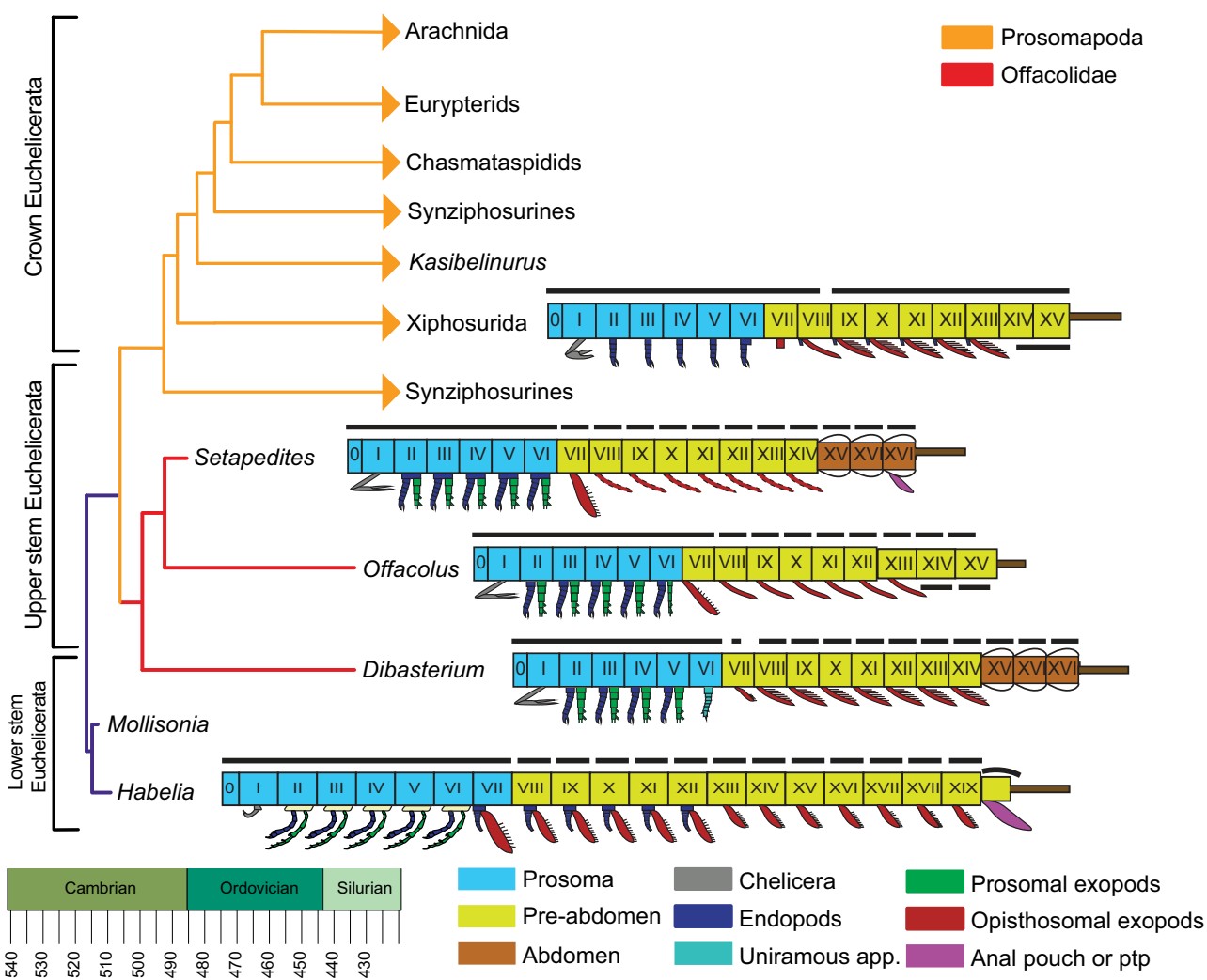

**Fig. 6 | Phylogenetic position of *Setapedites abundantis* gen. et sp. nov. among panchelicerates, showing early euchelicerate body plan evolution.** Simplified extended majority rule tree of a Bayesian analysis chronogram of euchelicerate relationships, based on a matrix of 39 taxa and 114 discrete characters, showing the position of *Setapedites abundantis* within Offacolidae. Lineages extending after the Silurian are indicated with arrowheads. Schematic models of the body organization in *Habelia*, *Setapedites abundantis*, *Dibasterium*, *Offacolus*, and Xiphosurida illustrate the origin and early evolution of euchelicerate uniramous prosomal appendages and tagmosis. Roman numbers designate somites. Prosoma somites are highlighted in blue, pre-abdomen somites in yellow, abdomen somites in brown, and the possible anal pouch or post-ventral structure (pvs) in purple. Black dorsal lines indicate tergites and cephalotorax. Schematic model of Xiphosurida *Offacolus*, and *Dibasterium* from[45], *Habelia* based on description from[12]. See Supplementary Fig. 6 for the full phylogenetic tree and Supplementary Data for phylogenetic characters and matrix.

---

*abundantis* supports the hypothesis that the divided biramous limbs of *Offacolus kingi* and *Dibasterium durgae* evolved from biramous appendages with a common limb base[6,12,17,43].

**Tagmosis, the seventh somite and ancestral reconstruction of the euchelicerate ground plan**

*Setapedites abundantis* possesses a seventh somite with a distinct tergite and appendages with a morphology similar to the opisthosomal appendages. Within Offacolidae, *Dibasterium* has a seventh somite bearing a micro-tergite and appendages with a reduced distinct morphology (unlike either the prosomal or opisthosomal appendages), and in *Offacolus* the seventh somite lacks a distinct tergite and bears appendages similar to those of the opisthosoma (lack of endopods and flat exopods). Likewise, *Habelia* has also been described as possessing a seventh somite lacking a distinct tergite and bearing appendages with morphology similar to the opisthosomal appendages. *Offacolus* and *Habelia* also both have a prosomal dorsal shield covering the first seven pairs of appendages. Despite these similar conditions, Dunlop &

Lamsdell[45] interpret the prosoma of *Offacolus* as having six somites on the basis of the morphology of the appendages, whereas Aria & Caron[12] interpret the prosoma of *Habelia* as having seven somites on the basis of the cephalic shield covering and shared specialized feeding function of the first seven appendage pairs despite their different morphology.

Discussions of ancestral reconstructions of the euchelicerate ground plan and the number of somites found in the prosoma and the opisthomsoma at the base of Euchelicerata require a clear definition of tagma versus pseudotagma (see Supplementary Information for additional discussion about the definition of tagma versus pseudotagma). In this manuscript, we follow Lamsdell[4] and Dunlop & Lamsdell[45] in distinguishing between a tagma as a region of functional specialization predominantly identified by modification or suppression of appendages, whereas a pseudotagma shows differentiation in the tergites or sternites without associated change in the form or function of appendages. In this framework, *Setapedites* is important because it is a stem lineage euchelicerate with a seventh somite that has a distinct and well-defined separate tergite and an opisthosomal-

like appendage (lack of endopods and flat exopods) associated with it, such that the seventh somite cannot belong to the prosoma. *Setapedites*, and probably *Dibasterium* with its micro-tergite, have a cephalic pseudotagma that matches the cephalic tagma, with both six tergites incorporated into the shield overlapping six pairs of appendages, whereas *Offacolus* has seven tergites pseudotagma and six somites tagma. Regarding *Habelia*, what Aria & Caron[12] interpreted as a seven-somite prosoma (tagma) is what we interpret as seven tergite dorsal shield (pseudotagma), and we regard the seventh pair of appendages, in this basal taxon, as unclear in whether they are associated with the prosoma or opisthosomal. Our ancestral state reconstruction supports a seven tergite dorsal shield at the base of Euchelicerata, but not necessarily a seven-somite prosoma tagma, as stated by Aria & Caron[6,12] (see Supplementary Fig. 16 for the character history reconstruction of the cephalic pseudotagma and Supplementary Information for additional discussion about the definition of tagma versus pseudotagma). What we see for these stem lineage euchelicerate taxa is a myriad morphologies for the seventh somite and associated appendages, underlining the morphological plasticity of this segment at the boundary between two major body tagma.

Regarding the evolution of the seventh pair, and following opisthosomal, appendages, most of our phylogenetic analyses (Fig. 6, Supplementary Figs. 5, 6 and 8) retrieve the wide paddle-like morphology of the exopods of somite VII in *Setapedites abundantis* and *Offacolus kingi* as likely plesiomorphic for the Offacolidae, considering the similar morphology of those exopods in *Habelia optata*. Given that *Dibasterium durgae* possesses a reduced condition of the exopod of somite VII[17], it has been considered as homologous to the xiphosurid chilaria[46] and the metastoma of eurypterids and chasmataspidids[27]. While we agree on the homology between those somites and related appendages, the peculiar morphology of the seventh pair of appendages in *Dibasterium* is more likely autoapomorphic, if considered into the broader Offacolidae array of morphologies for these appendages. The more posterior opisthosomal appendages in *Dibasterium durgae* and *Offacolus kingi* are modified into gill opercula, but this condition cannot be clearly stated for *Setapedites abundantis* (Figs. 3 and 6, Supplementary Fig. 1G and 1H), which exhibits a similar proximal portion of the opisthosomal appendages to that of *Habelia optata* (Figs. 3 and 6). While a bilobed paddle-like morphology of the opisthosomal exopods of *Setapedites abundantis* implies an independent evolution of lamellate gills in Offacolidae and crown-Euchelicerata, it will also support the notion that euchelicerate book gills evolved from *Habelia optata* bilobate paddle-like exopods[6]. The most parsimonious interpretation for this character is however, for it to represent a closer anatomy with other offacolids (see Supplementary Fig. 17 for the character history reconstruction of the opercula). Consequently, gill opercula can still be considered apomorphic of euchelicerates. Based on the anatomy of *Setapedites abundantis*, the oldest euchelicerate described, we suggest the body tagmatization into a prosoma composed of six somites (bearing the chelicerae and 5 pairs of appendages), pre-abdomen (bearing gill opercula), and abdomen, as diagnostic for Euchelicerata[4] in opposition to the gill operculum as the unique synapomorphy of Euchelicerata[6,45,47].

## Methods

### Experimental model and subject details
All material of *Setapedites abundantis* used in this study comes from the Fezouata Shale (Early Lower Ordovician, late Tremadocian) in the Zagora province, (Morocco) and was collected by authorized and academically recognized avocational Moroccan collector Mohamed Ben Moula and his family over the period of 2015 to 2016 (MGL collection), and between 2009 to 2014 (YPM collection); ACD together with paleontologist Peter Van Roy worked in collaboration with them to collect the metadata associated with the collected fossils.

Mohamed Ben Moula has a long-standing working relationship with several academics, has received the Mary Anning Award from the Paleontological Association, and has a radiodont fossil named after him by the same two authors (Van Roy et al. 2015b) in honor of his great contribution to the field of paleontology. The MGL fossil collection was purchased with funds from the University of Lausanne and the Swiss National Science Foundation, following all regulations for purchases. The fossil collection was transported to Casablanca and subjected to export approval by the Ministry of Energy, Mines and the Environment of the Kingdom of Morocco and approved for shipment to Switzerland on 11.05.2017 (export permits curated with the collection). Fossils were shipped by sea and land to the University of Lausanne, where they are curated as part of the collection of the Musée Cantonal de Géologie (MGL). The collections of the Yale Peabody Museum of Natural History were obtained both through a collection of specimens by Peter Van Roy during fieldwork and through purchase using dedicated museum funds for the acquisition of scientific collections. Export permits were obtained through the Moroccan Ministry of Energy, Mines, and the Environment, with specimens being transported from Casablanca by sea.

This published work and the nomenclatural acts it contains have been registered in ZooBank, the proposed online registration system for the International Code of Zoological Nomenclature (ICZN). The ZooBank LSIDs (Life Science Identifiers) can be resolved, and the associated information viewed through any standard web browser by appending the LSID to the prefix "http://zoobank.org/". The LSIDs for this publication are: *Setapedites abundantis* Lustri, Gueriau & Daley, In Press, LSIDurn:lsid:zoobank.org:act:A122B462-E673-4F8C-8F03-74C8905F7D63.

### Fossil photography
Specimens of *Setapedites abundantis* were initially examined with a light microscope (Wild Heerbrugg M8) under full-spectrum light. Subsequently, selected specimens were photographed with a digital SLR camera (Canon EOS 800D equipped with CANON MACRO LENS MP-E 65 mm 1:2.8 1-5X) mounted on a stand and connected to a z-stacking system (STACKSHOT 3X), using different lighting conditions: normal or polarized light, dry or covered in alcohol. Z-stacks were rendered using Helicon Focus software and the depth map function. All specimens were analysed in the Optical laboratory at Lausanne University (GEO-3439).

### Interpretative drawings
Interpretative drawings were made in Adobe Illustrator, using a Wacom ver. 6.3.38-3 graphic table and a pen tool on the photos.

### Synchrotron tomography and 3D rendering
Two specimens (MGL.102637a and MGL.102841) were imaged using synchrotron X-ray microtomography at the X02DA TOMCAT beamline of the Swiss Light Source, Paul Scherrer Institut, Villigen, Switzerland. Measurements were performed using a monochromatic beam of 35 and 18 keV respectively, a single propagation distance of 250 mm, a 100 µm LuAg:Ce scintillator, and a 4× objective, yielding reconstructed tomographic data with a voxel size of 1.75 µm. 1501 projections were recorded over 180° with exposure of 400 and 1000 ms, respectively. Reconstruction was performed on a 60-core Linux PC farm using a Fourier transform routine and a regridding procedure[48]. Virtual sections presented in Fig. 4 were processed using ImageJ, and segmentation and three-dimensional rendering was performed using the software MIMICS Innovation Suite 19.0 (Materialize) at the IPANEMA laboratory (Gif-sur-Yvette, France).

### Chronogram
Chronological data for the 39 taxa included in the parsimony analyses were collected from PBDB (paleobiodb.org)[49] and from data available

in[4], and are available in Supplementary Table 3. The tree resulting from Bayesian phylogenetic analyses was plotted against the geological timescale using the *strap* R package[50] in RStudio[51] in order to obtain the chronograms presented in Fig. 6 and Supplementary Fig. 6. Tree was rescaled with command "mbl".

## Phylogenetic analyses

The first Bayesian phylogenetic analyses (Supplementary Fig. 5, 9 and 10) were performed following methods in Aria & Caron[6], using mrBayes ver. 3.2.7a[52,53], through the Cipres science gateway web site (www.phylo.org)[54] (modifications and character coding for *S. abundantis* are available in Supplementary Information). Tree searches followed an Mkv + Γ model (Lewis 2001) with four chains sampling during four runs for 10,000,000 Markov chain Monte Carlo generations, a tree sampled every 1000 generations and burn-in of 20%. Analyses were constrained with a partial backbone. The character matrix was based on that found in Aria & Caron[6], with our final data matrix including 102 taxa and 267 discrete characters. Two versions of this matrix have been analysed to test two different character codings for *Mollisonia plenovenatrix* (see Supplementary Notes for a list of modified codings and Supplementary Data 1 and 2 for phylogenetic characters and matrix). A constraint for the Arachnopulmonata has also been used for the analysis resulting in Supplementary Fig. 9 (data available in Supplementary Data 3 for phylogenetic characters and matrix).

The second set of Bayesian phylogenetic analyses (Fig. 6 and Supplementary Fig. 6 and 11) were performed on the data matrix from Lamsdell[4], modified by removing *Fuxianhuia protensa* and *Willwerathia laticeps* from the data matrix (given its uncertain status as basal euarthropod in the first case[55] and as euchelicerates in the second[56,57]), using *Yohoia tenuis* as outgroup, and adding *Setapedites abundantis*, *Dibasterium durgae* (based on[3,17]), *Habelia optata* (based on[12]), and *Mollisonia plenovenatrix* (based on[6]). Two versions of this matrix have been analysed to test different characters codings for *Mollisonia plenovenatrix* (see Supplementary Notes for list of modified codings and Supplementary Data 4 and 5 for phylogenetic characters and matrix). Modifications and character coding for *Setapedites abundantis*, *Dibasterium durgae*, *Habelia optata* and *Mollisonia plenovenatrix* are available in Supplementary Information and Data. The final data matrix includes 40 taxa and 114 discrete characters. We followed the same methods of the previous Bayesian analyses but without partial backbone constraint.

The first parsimony analyses (Supplementary Fig. 7, Supplementary Data 6) were performed on the same data matrix modified from Lamsdell[4] and following its methods using TNT ver. 1.5[58] (random addition sequences followed by branch swapping with 100,000 repetitions, all characters unordered and of equal weight, followed by jacknife (33% deletion, 1000 repetitions) and bootstrap (50% deletion, 1000 repetitions)).

The second set of parsimony analyses (Supplementary Fig. 8, Supplementary Data 6) was performed on the same data matrix modified from Lamsdell[4] and following its methods using TNT ver. 1.5 except an implied weight of 12 K (data available in Supplementary Data 2 for phylogenetic characters and matrix). The exclusion of *Fuxianhuia protensa* from these analyses and the usage of *Yohoia tenuis* instead, lead to a paraphyly of the megacheiran. To test if this negatively affected our results in the other reiteration, we ran another set of phylogenetic analyses excluding the artipodans (*Olenoides serratus*, *Emeraldella brocki*, and *Sidneyia inexpectans*; Supplementary Fig. 12–15, Supplementary Data 7–9, see also Supplementary Table 2 and supplementary discussions regarding the different phylogenetic approach).

## Data availability

The fossils studied herein are curated as part of the collections of the Musée cantonal de géologie Lausanne (MGL), Switzerland, and the Yale Peabody Museum of Natural History (YPM), New Haven, CT, USA. Chronological data are available as Supplementary Table 3 in the Supplementary Information file. The SWISSUbase repository holds phylogenetic matrices (Data→ data_used_in_the_manuscript→ phylogenetic_analyses, file names ending in SD1 through SD9) as well as all photographs and tomograms (https://doi.org/10.48657/4whn-ak94).

## Code availability

The only codes associated to this work, used to run the phylogenetic analyzes and the chronological scaling of the tree, are available on the SWISSUbase repository (https://doi.org/10.48657/4whn-ak94).

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

## Acknowledgements

Specimens were collected and made available by the Ben Moula family, who also supported the fieldwork of the authors. We thank Susan Butts and Jessica Utrup for providing access to the specimens hosted at the Yale Peabody Museum and Antoine Pictet, Robin Marchant, and Lukáš Laibl for assistance with the collection of the Musée cantonal de géologie of Lausanne. We thank Peter Van Roy for his collaboration on this research including providing a crucial specimen, participating in field locality data collection, in-depth discussions of the fossils, and edits on the manuscript at different stages. We thank Russell J. Garwood for revising this manuscript in an early stage as part of Lorenzo Lustri's PhD thesis examination. We are grateful to Javier Ortega-Hernández, Rudy Lerosey-Aubril, David Sinclair, and Francesc Pérez-Peris for discussions,

to Oksana Vernygora for help with the phylogenetic analyses, to Lukáš Laibl for locality information, and to Elissa Sorojsrisom for the reconstruction presented in Fig. 5. Imran Rahman and Duncan Murdock provided access to the X02DA TOMCAT beamline at the Swiss Light Source, and we thank Christian Schlepütz and Federica Marone for their assistance at the beamline. L.L., P.G. and this research were funded by the Swiss National Science Foundation, grant number 205321_179084 entitled "Arthropod Evolution during the Ordovician Radiation: Insights from the Fezouata Biota" and awarded to A.C.D.

## Author contributions

L.L. photographed and drew the fossils, performed the phylogenetic analyses, prepared the figures, and wrote the manuscript under the supervision and editing of P.G. and A.C.D. P.G. performed the tomographic acquisitions, processed the data, and assisted/performed fossil photography and imaging.

## Competing interests

The authors declare no competing interests.
