## [Peer Review File · Nature Communications]

Lower Ordovician synziphosurine reveals early euchelicerate diversity and evolutionReviewers' Comments:

Reviewer #1:

Remarks to the Author:

This is a welcome contribution featuring new fossil material from the Fezouata Lagerstätte which indeed builds onto our knowledge of early euchelicerate evolution. For those who are a bit familiar with its history, the collection of Fezouata material and its study is a major undertaking, faced with various challenges, and such a publication is much, much more than the sum of its words and figures. The piece is well presented, organized and written, the figures are overall of good quality (but please check your panel labelling before submission!) and the phylogenetic analyses were executed properly—although there are some small issues. However, I also see a number of shortcomings and omissions, for instance with the systematics and the presentation of the evidence.

The title is a bit over-reaching. As I develop below, the fossil morphotype arguably cements a taxon including a couple of horseshoe-crab-like genera from the Herefordshire biota, adding to morpho-anatomical diversity in this group, but does not exactly “reveal” the origin of euchelicerate appendages—also because there are Cambrian fossils that also contributed to build the picture of euchelicerate origins. A much more accurate (and no less impactful) title could be “An Ordovician fossil documents early euchelicerate diversity”, or something of the sort. In any case, quotation marks are unnecessary for synziphosurines because it’s a vernacular name (“Synziphosura” would require them).

In the Arachnoplumonata context, some colleagues question whether the choice of topological constraints in my matrix influence the placement of fossils in a certain way. I obviously support the basal placement of synziphosurines, and incidentally that of scorpions, but it would be good if you ran the analysis with an Arachnoplumonata constraint, just to see what you obtain—and whatever the result, this would warrant but a couple lines of discussion.

I am quite puzzled as to why you are not erecting a new family. Taxonomy requires it at this point. You would propose a new clade with no Linnean rank if you were describing a putative clade at the supra-familial or ordinal level maybe, but here, even if your clade is not very well supported, the grouping is justified. Accordingly, the family name should be Offacoliidae.

There are some issues with the phylogenetic treatment:

- Knowing that Bayesian analyses generally outperform parsimony, why use a Bayesian approach for one matrix but parsimony for the other? If you defend the value of parsimony, you should then run parsimony in both cases, although it might be tedious to implement the backbones. Run at least your second analysis with MrBayes.
- What is your outgroup in your analysis of the Lamsdell matrix? You did well to include Habelia and Dibasterium, but Mollisonia is crucially missing, and rooting on Fuxianhuia is ill-advised, given that its status as basal euarthropod is contested from several fronts. I tested a parsimony run with Yohioia as outgroup (your best candidate here) and Fuxianhuia resolves as sister-group to Panchelicerata. Something’s off. You might as well remove it entirely.
- For the sake of reproducibility, please indicate whether you used sectorial search, ratchet, drift and tree fusing, and with what settings.

The absence of Mollisonia throughout is also something I find very strange, especially since you say you reused the data from that paper. If you disagree with the interpretations, state it somewhere, but I don’t see why, because it wholly integrates the evolutionary picture you’re describing.

Taxonomy: Besides the family issue, I find the name wanting. Any stem taxon to an extent could be a chimaera of what it stands between, so the genus name doesn’t say much. My recommendation would be to take the opportunity to put forward the Fezouata locality and its people and culture. As for the species name, “robustus, a, um” and “axis, is” are indeed Latin words (and should be referred to), so it works technically, but “axis, is” being a masculine substantive (3rd declension), you are in fact

putting the species name as a masculine apposition of a feminine... It's odd. "robusta" could be a species name while transferring the "axis" idea to the genus. In any case, there's a lot of room to be more creative here!

The diagnosis of "Setopoda"... Biramous prosomal appendages with "detached" stenopodous exopods are plesiomorphic since they are present in habeliidans already, so this shouldn't be part of your diagnosis. A "brush" tip is also present in Mollisonia, and I draw your attention to the fact that there is a confounding ramus with the same termination in Sanctacaris (Aria and Caron 2017 Add. File 8)—which makes "Setopoda" not very fitting, but see my comment about erecting a family instead anyway. As unambiguous diagnostic traits, when we consider Offacolus and Dibasterium for now, this leaves the bifurcating telson tip, the lobate and setose seventh prosomal appendage, but also the elongate chelicerae and, importantly, the extensive reduction of gnathobasipods.

Now, these are nice specimens, but they are not very detailed either, and the preservation is challenging:

- Given the arrangement of the matrix, I am skeptical about a clear bifurcation of the telson tip in MGL.102690. I expect it's very likely to be the case, but we have to be honest with the evidence at hand. Try to make a better case for it by adding a close-up in Supplementary.
- Setation is not preserved, and seeing the identity of the seventh prosomal appendage compared to opisthosomal ones is frankly a leap of faith. Provide close-ups of these relevant specimens in Figure 2 for the proso-opisthosoma transition.
- Yes, the axial area is well defined, but is it so different from what is known in Offacolus and Dibasterium? Remember that we are comparing compression fossils with fully 3D nodular ones... The tergite-pleurae demarcation is lesser in dorso-oblique preserved specimens.
- You interpret your opisthosomal appendages as vaguely "paddle-like", implying they are not operculate book-gills, and therefore that your taxon is not a euchelicerate (contradicting your sys pal), and also that this diagnostic feature might have originated independently in Offacolus + Dibasterium and other euchelicerates... This begs caution (see also my comment below). The fact is that indeed your opisthosomal appendages are not well preserved, and it is entirely possible that you have book gills like in the sister taxa. If, after coding these states as "?" in your matrix, you still retrieve your "Setopoda" clade, then you can invoke secondary homology and justify assuming operculate book gills. Otherwise, your taxa should more likely be paraphyletic without this feature.
- I do not entirely agree with your assessment of the first opisthosomal segment. It appears to me differentiated from posterior segments, larger notably, and overlap by the cephalon can be extensive. In that sense, this segment is not so different from what is observed in Dibasterium.

This really leaves for the diagnosis of Offacoliidae;

- The elongate chelicerae: In your morphotype they are shorter than in Offacolus or Dibasterium (which might support their grouping), but elongate also compared to stem euchelicerates or synziph. There's of course variability in eurypterids and arachnids but local apomorphies are just a Cambrian reality (and why Bayesian phylogenetic treatments are better);
- The small size of the gnathobasipods;
- The morphology of the seventh prosomal appendage;
- Maybe the telson tip.

It is therefore essential to me that you be more circumspect with this fossil evidence and that you tighten up the taxonomic treatment, as well as a number of other aspects, as detailed below.

See also below about making your main figures more useful.

Other comments:

41 – The text here and there could do with more punctuation. "and their closest relatives, the xiphosurans..."

43 – This sentence is not accurate. Euchelicerata is diagnosed by the one synapomorphy that is the

presence of book gill opercula. Both chelicerae and tagmatization are present in the stem in the strict sense (see Aria and Caron 2019 Nature).

48 – This presentation is somewhat outdated. Since the publication pertaining to habeliidans and Mollisonia, there haven't been counter-arguments as to how megacheirans or some artiopodans could be closer to Euchelicerata. The rooting of Euchelicerata by Mollisonia and habeliidans was retrieved independently by Zeng et al. 2020 (Nature). The issue I think is more about how megacheirans or certain artiopodans articulate, if at all, the Panchelicerata clade that would start with habeliidans.

53 – It is "habeliidans" and not "habeliids" – "habeliids" would refer to the family Habeliidae, but here, it is the order Habeliida. This was coined by Aria and Caron 2017.

54 – I think you mean the exopods of xenopodans (Emeraldella, Sidneyia). The trunk appendages of Sanctacaris like that of Habelia are broad and lobate and their proximalmost portion is not well known. The argument of Legg was more a comparison between the head of Sanctacaris and that of xenopodans and then with the exopods of xenopodans and those of xiphosurans. However, the interpretation of the head was, like the original one, not entirely correct.

Also, please choose to use exite or exopod throughout your work, for consistency, and make sure to explain your choice if you choose exite.

58 – Given how Offacolus and Dibasterium are the central, most relevant taxa to your study, you should introduce them a bit more, and especially say how they have consensually been retrieved as basalmost euchelicerates across studies. Say more about their main morpho-anatomy in introduction.

69 – "This species displays..." This last part repeats the abstract and is not necessary.

135 – That's indeed intriguing.

140 – "Description" should be continuous with the systematic palaeontology (italicized).

243 – I don't think "doubts" can be "enhanced"...

245 – Again, there wasn't exactly a debate... The matrix by Legg hasn't been published with updated information about habeliidans and Mollisonia recently, as far as I'm aware, so this is not a matter of whether, in this specific case, there were two sets of data with opposite signals. Like I wrote before, it is more about 1) a historical reference to the open question whether megacheirans, "vicissicaudates" or another group represented the closest sister group to Chelicerata, which translated into 2) how Megacheira and Artiopoda articulate with Panchelicerata. In that sense, the significance of your taxon, in my view, is more that it *finally bridges the gap with these Silurian horseshoe-crab-like forms and stem euchelicerates from the Cambrian explosion*.

248 – I don't understand what you mean here. And yes, be careful that the phylogenetic resolution of sea spiders is a long-standing problem, and unless you're extremely confident about your branch support, I'd refrain from making such a statement.

252 – No. The gill operculum is the unique synapomorphy of Euchelicerata (as per Weygolt and Paulus; see also e.g. Dunlop 1997, Dunlop and Lamsdell 2017, Aria and Caron 2019), and unless you're willing to amend the taxonomy (which I would *not* recommend doing because there is no need to), you have to remove this statement.

253 – 254 – Dibasterium and Offacolus were considered euchelicerates before, and were already forming a clade basal to "Prosomapoda." And obviously Euchelicerata was already monophyletic. You should focus on the novelties.

255 – As explained above, you cannot be certain.

257 – Again, refer to the actual diagnosis of Euchelicerata.

260 – A number of these characters are actually shared with Sanctacaris. Why is it absent?

269 – To be clear, you construe that the megacheiran-like exopods of the seventh pair of cephalic appendages in Habelia are homologous to the seventh pair of appendages in "setopods", implying the reduction of the endopod.

275 – I think it's fair to cite Aria and Caron 2019 Nature here.

285 – Again, see Mollisonia, specifically ROMIP 65262. The taxon has the same "brush" terminations in the prosoma. The absence of the feature in habeliids just has to do with the fact that the exopods are very slender (like exopod #6 in Offacolus), with small tips, but a bunch of setae is present too.

290 – You identify what possibly is the basipod of an appendage (as was done with Offacolus and Dibasterium already), but nowhere I see that you can demonstrate where exopods attach. Given the resolution of your material compared to the Burgess Shale or the Herefordshire, I think this will be

difficult.

291 – Proximalmost appendage morphology in Habeliidans as well as Mollisonia is no less clear than in Offacolus and Dibasterium: in all cases, the semi-independence of the exopod is based on the absence of evidence for the point of attachment and the observation that they do not seem constrained by the movement of the endopods. Which means that this condition started off with Panchelicerata. Thus the polarization is quite clear. Did this lead to the loss of prosomal exopods? Yes, most likely – this was discussed before (Aria and Caron 2017, 2019, Aria 2021 PaleorXiv).

294 – What about exite instead of exopod?

306 – Mmmm... but 1) nothing here is new, because this scenario was discussed in the context of Mollisonia, and 2) the morphology of opisthosomal limbs in your fossil is unclear beyond a lobate aspect (that could very well be opercula!).

307 – Whether the name should define the group of taxa or the reverse is always a tautological argument. If you say “Should Euchelicerata represent the least inclusive clade containing XXX”, then I understand, but this is explicitly at odds with the current formal diagnosis. I do see the issue that the absence of opercula in your taxon could render Euchelicerata polyphyletic, but 1) like I wrote above, the presence of opercula remains entirely possible, and 2) this is dependent on the topology and the robustness of the results. As a matter of fact, if you support the monophyly of “setopods” and its polytomy, the presumed absence of opercula in your morphotype would parsimoniously optimize as a reversal (ACCTRAN or DELTRAN), contradicting your statement. Why don’t you optimize important character state changes on your topology? This would make it very clear.

Figures: Some of your Supp. figure panels are much more interesting to put as main figure panels, e.g. 2C or E (but not 2F) (that synchrotron effort should have more visibility!), 3A-B (for the chelicerae and putative bipartite epistome-labrum), a larger version of 3J. In your main figures 1 and 2, many panels are redundant or don’t show much, so they can easily be replaced.

Figure 1. L panel should be K.

Figure 2. In E, please correct tone – this is much too yellow and hampers visibility of features.

Figure 2C. If the labrum is the posterior lobe of the structure you divide in 2, what’s the anterior portion? The epistome? This is a bit of a rhetoric question, as you’ve probably reached the limit of informative resolution here. I think you’re better off calling the entire area putative epistome-labrum (if you want to keep with chelicerate terminology, or any other term as long as it’s justified).

Supplementary Figure 3. Drawings G and O are overinterpretations. You draw all boundaries as if those were genuine, but it is evident that the structures are severely altered and that you cannot make the case for these extra long spines, etc.

Supplementary Figure 4. Put the PPs everywhere, not just for the Chelicerata clade.

Supp. discussion – Prosoma ground pattern: Habeliidans and mollisoniids, as well as some other unpublished Burgess Shale taxa, have seven pairs of appendages under the cephalic shield. Should these taxa form a clade, this could be indeed autapomorphic. But it is also not unfathomable to think that this tergite was secondarily released—this is commonplace among crustacean cephalothoraxes! I don’t think there is a need for you to oppose the evidence so strongly, especially when it’s otherwise very clear in many specimens and several taxa.

I hope these comments were fair and constructive. I will be glad to see a revised version of this work and, if satisfying, recommend it for publication.

Cédric Aria

Reviewer #2:

Remarks to the Author:

This is a nice manuscript describing a very common synziphosurine from the Fezouata biota. The record is important for a number of reasons in terms of both its age and the quality of preservation.

The manuscript is well written and well presented. I am not a specialist on chelicerates so I decided to simply read the manuscript and see if the presentation was sound and whether I could identify the characters in question.

The specimens are really small. Hence the appendages are faint infill of iron oxides and clays after pyrite. I bet some referees may say they can't see the podomere boundaries and while the images are not always the best ones I can see them and believe the authors' interpretation. So, in case you get some obstinate referees saying that they can't see a thing, tell them that its because they are blind or at least not in the possession of trained eyes to look at exceptional fossils and hence should abstain from commenting on such issues again.

I can see the biramous prosomal appendages and the chelicerae. I am struggling a bit more with the opisthosomal appendages and their morphology, but agree that there is some structures that clearly derives from exposure of structures below the tergal skeleton and which is appendicular in nature. The supplemental figures are more helpful.

Too bad these couldn't have been revealed with the synchrotron. I surely would have taken 20-40 individuals to the machine after having cut them down to size and scanned the hell out of them. Then I would have manually segmented them out with a nice bottle of Swizz white wine by my side rather than thresholding them which often miss more faintly preserved features. Surely a lot of cool stuff would be revealed that way and some specimens could reveal more appendicular details, but given the approaches done here, this manuscript is sufficient for the description to be useful for other researchers, so I will not linger on this thought or suggestion for longer.

I notice that the material is supposedly all based on collections made in 2015 and 2016 and kept in both Lausanne and Yale. Now, I believe a lot of the material at Yale was collected in 2009 and up to 2011/2012. I was involved in collecting some of this material with PVR and I remember specimens of similar quality as the ones shown here and I even have a couple of specimens in Bristol preserving nice limbs. I wonder why this material was not included or mentioned?

Minor additional comments.

Figure 2 labels are confused. I was looking for the opisthosoma appendagel panels (Fig. 2E,F), but the labels indicating those panels are missing and then the subsequent panel labels follows a much higher set of alphabetical letters. I guess some editing between journals led to this omission.

I don't see any reason to hold this manuscript up from what I can gather. Unless there is an issue with the taxonomic work, which someone like David Legg, Greg Edgecombe, Jason Dunlop, James Lamsdell or such noticed I think it is fine to publish with a few tidbits to be clarified. I don't need to see the manuscript again.

Best

Jakob Vinther

Reviewer #3:

Remarks to the Author:

What are the noteworthy results?

This is the first formal description of this common element of the Fezouata biota, a very important Ordovician Fossil-Lagerstätte. Comparison of other early Palaeozoic euclerates with appendages revealed that the new genus provides important information concerning the evolution of euclerate appendages.

Will the work be of significance to the field and related fields? How does it compare to the established literature? If the work is not original, please provide relevant references.

The work is significant because it adds to our knowledge of early euclerates.

Does the work support the conclusions and claims, or is additional evidence needed?

In general, but the new genus has only been compared, at least in the text (the supplementary material is simply NEXUS files), to other synziphosurines with appendages.

Are there any flaws in the data analysis, interpretation and conclusions? Do these prohibit publication or require revision?

The main problem with this work is that, at first sight, the new genus looks remarkably like other synziphosurines, such as *Pasternakevia* Selden & Drygant, 1987 (poorly redescribed by Krzemiński et al. 2010) and aglaspidids such as *Tremaglaspidis* Rushton & Fortey (see, for example, Lerosey-Aubril et al. 2013), and others. Most of these do not show appendages, but this is no reason to exclude them from the analysis. How do we know they are unrelated to the new genus? I would like to see discussion of these genera, which greatly resemble the new form, at least in dorsal morphology.

Is the methodology sound? Does the work meet the expected standards in your field?

See comments above re comparison with other synziphosurines and aglaspidids.

Is there enough detail provided in the methods for the work to be reproduced?

Yes.

References

- Krzemiński, W., Krzemińska, E., and Wojciechowski, D. 2010. Silurian synziphosurine horseshoe crab *Pasternakevia* revisited. *Acta Palaeontologica Polonica* 55 (1): 133–139.
- Lerosey-Aubril, R., et al. 2013. "Occurrence of the Ordovician-type aglaspidid *Tremaglaspidis* in the Cambrian Weeks Formation (Utah, USA)." *Geological Magazine* 150(05): 945-951.
- Rushton, A. W. A. and R. A. Fortey 2003. "A new aglaspidid arthropod from the Lower Ordovician of Wales." *Palaeontology* 46(5): 1031-1038.
- Selden, P. and Drygant, D.M. 1987. A new Silurian xiphosuran from Podolia, Ukraine, USSR. *Palaeontology* 30: 537–542.

We thank all three reviewers for their highly beneficial and constructive comments, which we think improved the quality of our manuscript as a whole. In order to match the reviewer's suggestions we have renamed our fossil and we have improved our phylogenetic analyses, including a new taxon (*Mollisonia*) into the matrix, changed the outgroup and performed an analysis with Bayesian methodology for both the matrices. We have also restructured most of our figures in order to increase the quality of the pictures shown and we have added a new whole figure in the main text showing two different specimens that have been scanned with synchrotron X-ray microtomography and manually segmented. Edits to the discussion have been made in order to acknowledge properly different hypotheses regarding the early evolution the euchelicerates. We also include in the supplemental information two new specimens to better illustrate critical characters such as the bifurcate telson and the opisthosomal appendages as well as two whole new paragraphs treating the relationship of our new taxa with other synziphosurines and Aglaspids and the interpretation of its labrum.

Reviewer #1 (Remarks to the Author):

This is a welcome contribution featuring new fossil material from the Fezouata Lagerstätte which indeed builds onto our knowledge of early euchelicerate evolution. For those who are a bit familiar with its history, the collection of Fezouata material and its study is a major undertaking, faced with various challenges, and such a publication is much, much more than the sum of its words and figures. The piece is well presented, organized and written, the figures are overall of good quality (but please check your panel labelling before submission!) and the phylogenetic analyses were executed properly—although there are some small issues. However, I also see a number of shortcomings and omissions, for instance with the systematics and the presentation of the evidence.

-We thank Cedric Aria for his supporting words and encouraging comments about this manuscript. It was indeed a major undertaking involving examining hundreds of fossils, and we are happy to hear that this work is appreciate by this reviewer. We have read his comments with great attention to detail, and have amended the manuscript to account for all of Cedric's excellent and constructive suggestions.

The title is a bit over-reaching. As I develop below, the fossil morphotype arguably cements a taxon including a couple of horseshoe-crab-like genera from the Herefordshire biota, adding to morpho-anatomical diversity in this group, but does not exactly “reveal” the origin of euchelicerate appendages—also because there are Cambrian fossils that also contributed to build the picture of euchelicerate origins. A much more accurate (and no less impactful) title could be “An Ordovician fossil documents early euchelicerate diversity”, or something of the sort. In any case, quotation marks are unnecessary for synziphosurines because it's a vernacular name (“Synziphosura” would require them).

-Amended. The title is now “Ordovician synziphosurine reveals early euchelicerate diversity and provides insight into euchelicerate appendage evolution”.

In the Arachnoplumonata context, some colleagues question whether the choice of topological constraints in my matrix influence the placement of fossils in a certain way. I obviously support the basal placement of synziphosurines, and incidentally that of scorpions, but it would be good if you ran the analysis with an Arachnoplumonata constraint, just to see what you obtain—and whatever the result, this would warrant but a couple lines of discussion.

-Amended. We tested the phylogeny using the Arachnoplumonata constraint and the result is attached in the material for the reviewer. However, we find this to be unrelated to the main story of our paper, so we decide to exclude it from the MS.

I am quite puzzled as to why you are not erecting a new family. Taxonomy requires it at this point. You would propose a new clade with no Linnean rank if you were describing a putative clade at the supra-familial or ordinal level maybe, but here, even if your clade is not very well supported, the grouping is justified. Accordingly, the family name should be Offacolidae.

-Amended. We include *Dibasterium durgae* and *Setapedites* in the Offacolidae.

There are some issues with the phylogenetic treatment:

- Knowing that Bayesian analyses generally outperform parsimony, why use a Bayesian approach for one matrix but parsimony for the other? If you defend the value of parsimony, you should then run parsimony in both cases, although it might be tedious to implement the backbones. Run at least your second analysis with MrBayes.

- We promote neither the specific merits of parsimony nor Bayesian analyses. We consider the most unbiased way to test phylogenetic hypotheses is to use both approaches since neither of them can be unequivocal in its results. We accepted the suggestion of using the Bayesian approach on the matrix from Lamsdell 2013, and as such, we replace the tree shown in figure 6 with the one obtained with the Bayesian analyses performed on it. We moved the tree obtained from the matrix from Lamsdell 2013 with maximum parsimony to the supplementary material.

- What is your outgroup in your analysis of the Lamsdell matrix? You did well to include *Habelia* and *Dibasterium*, but *Mollisonia* is crucially missing, and rooting on *Fuxianhuia* is ill-advised, given that its status as basal euarthropod is contested from several fronts. I tested a parsimony run with *Yohoia* as outgroup (your best candidate here) and *Fuxianhuia* resolves as sister-group to *Panchelicerata*. Something's off. You might as well remove it entirely.

-Amended. Thank you for the suggestions. We removed *Fuxianhuia*, and we now use *Yohoia* as outgroup. Also, we now include *Mollisonia* even in the analyses performed on the matrix from Lamsdell 2013.

- For the sake of reproducibility, please indicate whether you used sectorial search, ratchet, drift and tree fusing, and with what settings.

-Amended. We included the script used for the parsimony analyses on TNT command line version as a new supplemental file. This indicates the settings requested.

The absence of *Mollisonia* throughout is also something I find very strange, especially since you say you reused the data from that paper. If you disagree with the interpretations, state it somewhere, but I don't see why, because it wholly integrates the evolutionary picture you're describing.

-Amended. While we find the relationship of *Habelia* with euechelicerales strongly supported, we find the interpretation of *Mollisonia* as a euechelicerales with chelicera and proto-book gills less compelling. The preservation state of the chelicera is not optimal and could be interpreted differently (Budd, 2021, *Arthropod structure & development*, 62, 101048.), as well as the interpretation of the proto-book gills. Also, other papers such as Lerosey-Aubril et al. (2020, *PeerJ*, 8, e8879), and more recently description of the nervous system (Ortega-Hernández et al. 2022, *Nature Communications*, 13(1), 1-11.) suggested different affinities for *Mollisonia*. We did update the MS with an addition of this interpretation of *Mollisonia* in the first paragraph of the Discussion, but we did not move the focus of our comparison from *Habelia*. The addition of *Mollisonia* to all the phylogenetic analyses has been beneficial and it solidifies the status of Offacolidae as a group including *Offacolus*, *Dibasterium* and *Setapedites*.

Taxonomy: Besides the family issue, I find the name wanting. Any stem taxon to an extent could be a chimaera of what it stands between, so the genus name doesn't say much. My recommendation would be to take the opportunity to put forward the Fezouata locality and its people and culture. As for the species name, "robustus, a, um" and "axis, is" are indeed Latin words (and should be referred to), so it works technically, but "axis, is" being a masculine substantive (3rd declension), you are in fact putting the species name as a masculine apposition of a feminine... It's odd. "robusta" could be a species name while transferring the "axis" idea to the genus. In any case, there's a lot of room to be more creative here!

-Amended. We have changed the name to *Setapedites abundantis*.

The diagnosis of "Setopoda".... Biramous prosomal appendages with "detached" stenopodous exopods are plesiomorphic since they are present in habeliidans already, so this shouldn't be part of your diagnosis. A "brush" tip is also present in Mollisonia, and I draw your attention to the fact that there is a confounding ramus with the same termination in Sanctacaris (Aria and Caron 2017 Add. File 8)—which makes "Setopoda" not very fitting, but see my comment about erecting a family instead anyway. As unambiguous diagnostic traits, when we consider Offacolus and Dibasterium for now, this leaves the bifurcating telson tip, the lobate and setose seventh prosomal appendage, but also the elongate chelicerae and, importantly, the extensive reduction of gnathobasipods.

- Amended. In our interpretation of the phylogenetic results, Euchelicerata is a valid clade which does not include (at the present time) unequivocally *Mollisonia*, *Sanctacaris* or *Habelia*. As such, "euchelicerates with at least second to fifth prosomal appendage pairs being biramous and composed of stenopodous exopods of six podomeres bearing a brush-like group of setae on the distalmost podomere" is unique inside the monophyletic group of euchelicerata. Also, while it is certainly true these characters (biramous prosomal appendages with detached stenopodous exopods) are shared with habeliids (and we do acknowledge it, focusing much of our discussion on it) the presence of biramous prosomal appendages with detached stenopodous exopods associated with six podomeres bearing a brush-like group of setae on the distalmost podomere is unique of Offacoliidae. We are skeptical about the presence of homologous brush tips in the exopods of Mollisonia. However, we do now include elongate chelicerae and reduced gnathobasipods as a valuable addition to the amended diagnosis for Offacoliidae.

Now, these are nice specimens, but they are not very detailed either, and the preservation is challenging:

- Given the arrangement of the matrix, I am skeptical about a clear bifurcation of the telson tip in MGL.102690. I expect it's very likely to be the case, but we have to be honest with the evidence at hand. Try to make a better case for it by adding a close-up in Supplementary.

-We added images to show the bifurcation of the telson tip more clearly. We add images of a new specimen 102899 counterpart, with close-up available in supplementary material (supp. Fig. 4).

- Setation is not preserved, and seeing the identity of the seventh prosomal appendage compared to opisthosomal ones is frankly a leap of faith. Provide close-ups of these relevant specimens in Figure 2 for the proso-opisthosoma transition.

- We add a new specimen 102469 in the supplementary discussion and refine the contrast of the specimen in figure 2 (now figure 3), to show this anatomical feature more clearly.

- Yes, the axial area is well defined, but is it so different from what is known in Offacolus and Dibasterium? Remember that we are comparing compression fossils with fully 3D nodular ones... The tergite-pleurae demarcation is lesser in dorso-oblique preserved specimens.

-Amended. We remove from the description “strongly developed axis” and let only “developed axis”. Even if a certain degree of compression affects the preservation of the described fossils flattening them, they are not completely flat. The axis area is in all the specimens the most raised part of the body.

- You interpret your opisthosomal appendages as vaguely “paddle-like”, implying they are not operculate book-gills, and therefore that your taxon is not a euchelicerate (contradicting your sys pal), and also that this diagnostic feature might have originated independently in *Offacolus* + *Dibasterium* and other euchelicerates... This begs caution (see also my comment below). The fact is that indeed your opisthosomal appendages are not well preserved, and it is entirely possible that you have book gills like in the sister taxa. If, after coding these states as “?” in your matrix, you still retrieve your “Setopoda” clade, then you can invoke secondary homology and justify assuming operculate book gills. Otherwise, your taxa should more likely be paraphyletic without this feature.

We acknowledge that the presence of *Habelia*-like exopods in *Setapedites* implies a convergent evolution of lamellate gills in *Offacoliidae* and crown-Euchelicerata. We insert a mention to this in the discussion. However, we found paddle-like appendages and no trace of operculate book gills, so we state what we have and we have not written any statement about what we could have. We did not contradict ourselves because we state “This suggests that euchelicerate book gills evolved from *Habelia*- or *Setapedites*-like bilobate paddle-like exopods. Consequently, gill opercula have been acquired during euchelicerate evolution, and as such are not an apomorphy for Euchelicerata, unlike the chelicera and body tagmatization that are characteristic of the entire clade, including its stem members.”

- I do not entirely agree with your assessment of the first opisthosomal segment. It appears to me differentiated from posterior segments, larger notably, and overlap by the cephalon can be extensive. In that sense, this segment is not so different from what is observed in *Dibasterium*.

First opisthosomal segment in *Dibasterium* is described as “narrow transversely and lacks pleurae”(Briggs et al. 2012). The fully expressed first tergite in *Setapedites* is not narrow transversely, in fact even though it can be extensively overlapped by the prosomal shield it is comparable in width and length to the following tergites.

This really leaves for the diagnosis of *Offacoliidae*;

- The elongate chelicerae: In your morphotype they are shorter than in *Offacolus* or *Dibasterium* (which might support their grouping), but elongate also compared to stem euchelicerates or synziph. There’s of course variability in eurypterids and arachnids but local apomorphies are just a Cambrian reality (and why Bayesian phylogenetic treatments are better);
- The small size of the gnathobasipods;
- The morphology of the seventh prosomal appendage;
- Maybe the telson tip.

-We thank the reviewer for these comments that have helped us refine and clarify the diagnosis for *Offacoliidae*, for which we are grateful. The diagnosis now focuses on these features suggested by the reviewer and reads as follows:

Amended diagnosis : Euchelicerates with elongate chelicera, at least second to fifth prosomal appendage pairs being biramous and with reduced gnathobasipods and composed of stenopodous exopods of six podomeres bearing a brush-like group of setae on the distalmost podomere. Seventh pair of appendages

(sixth post-cheliceral appendage pair) uniramous and lobate paddle-like, fringed by lateral spines. Tip of the telson bifurcate.

It is therefore essential to me that you be more circumspect with this fossil evidence and that you tighten up the taxonomic treatment, as well as a number of other aspects, as detailed below.

We thank the reviewer again for these detailed comments, which greatly improved the taxonomic treatment of our new taxon.

See also below about making your main figures more useful.

Other comments:

41 – The text here and there could do with more punctuation. “and their closest relatives, the xiphosurans...”

-Amended.

43 – This sentence is not accurate. Euchelicerata is diagnosed by the one synapomorphy that is the presence of book gill opercula. Both chelicerae and tagmatization are present in the stem in the strict sense (see Aria and Caron 2019 Nature).

-The suggested single diagnostic character as used in Aria & Caron (2019) is in contrast to the more long-standing and generally accepted list of characters accepted by the community to define Euchelicerata. The count of body segments and tagmatization as well as the true nature of the supposed chelicerae of *Mollisonia* have already been questioned (Lerosey-Aubril et al. 2020. *PeerJ*, 8, e8879), with their interpretation being further weakened by recent information on the nervous system of *Mollisonia* (Ortega-Hernández et al. 2022. *Nature Communications*, 13(1), 1-11.) The formula we are suggesting is the least controversial and better represents our own views, as well as the more widely held views in the community, which is more reasonable for an introduction. We have however added reference to this hypothesis of one synapomorphy, along with the debate surrounding the hypothesis, to the manuscript, for example in the first paragraph of the Introduction.

48 – This presentation is somewhat outdated. Since the publication pertaining to habeliidans and *Mollisonia*, there haven't been counter-arguments as to how megacheirans or some artiopodans could be closer to Euchelicerata. The rooting of Euchelicerata by *Mollisonia* and habeliidans was retrieved independently by Zeng et al. 2020 (Nature). The issue I think is more about how megacheirans or certain artiopodans articulate, if at all, the Panchelicerata clade that would start with habeliidans.

-The different hypotheses regarding euchelicerate closest relatives cannot be solved by one single paper. We inserted a mention to *Mollisonia* later in the manuscript, in the first paragraph of the Discussion, to point out that it is one of the hypotheses, but we do not favor the interpretation of *Mollisonia* given in Aria and Caron (2019). As for the above *Mollisonia* interpretation as a stem-euchelicerates, this is not widely accepted and has been questioned already in the two recent publications mentioned in the previous paragraph.

53 – It is “habeliidans” and not “habeliids” – “habeliids” would refer to the family Habeliidae, but here, it is the order Habeliida. This was coined by Aria and Caron 2017.

-Amended. To avoid ambiguity, we amended the text to use the formal name of Order Habeliida throughout the manuscript.

54 – I think you mean the exopods of xenopodans (Emeraldella, Sidneyia). The trunk appendages of Sanctacaris like that of Habelia are broad and lobate and their proximalmost portion is not well known. The argument of Legg was more a comparison between the head of Sanctacaris and that of xenopodans and then with the exopods of xenopodans and those of xiphosurans. However, the interpretation of the head was, like the original one, not entirely correct.

-Amended. We deleted the reference to Sanctacaris.

Also, please choose to use exite or exopod throughout your work, for consistency, and make sure to explain your choice if you choose exite.

-Amended. Exopods is now used consistently throughout.

58 – Given how Offacolus and Dibasterium are the central, most relevant taxa to your study, you should introduce them a bit more, and especially say how they have consensually been retrieved as basalmost euchelicerates across studies. Say more about their main morpho-anatomy in introduction.

-Amended. A brief introduction to their morphology and their basal-euchelicerate affinity was added to the end of the first paragraph of the introduction.

69 – “This species displays...” This last part repeats the abstract and is not necessary.

-Amended. Repetitive text removed.

135 – That’s indeed intriguing.

We agree!

140 – “Description” should be continuous with the systematic palaeontology (italicized).

-Amended.

243 – I don’t think “doubts” can be “enhanced” ...

-Amended.

245 – Again, there wasn’t exactly a debate... The matrix by Legg hasn’t been published with updated information about habeliidans and Mollisonia recently, as far as I’m aware, so this is not a matter of whether, in this specific case, there were two sets of data with opposite signals. Like I wrote before, it is more about 1) a historical reference to the open question whether megacheirans, “vicissicaudates” or another group represented the closest sister group to Chelicerata, which translated into 2) how Megacheira and Artiopoda articulate with Panchelicerata. In that sense, the significance of your taxon, in my view, is more that it *finally bridges the gap with these Silurian horseshoe-crab-like forms and stem euchelicerates from the Cambrian explosion*.

Answered above.

248 – I don’t understand what you mean here. And yes, be careful that the phylogenetic resolution of sea spiders is a long-standing problem, and unless you’re extremely confident about your branch support, I’d refrain from making such a statement.

-Amended. We delete the phrase” The biramous prosomal appendages shared between *Habelia*, *Dibasterium*, *Offacolus* and now *C. robustaxis* support the hypothesis of a convergent origin for chelifores and chelicerae” being a controversial argument not at the center of our discussion.

252 – No. The gill operculum is the unique synapomorphy of Euchelicerata (as per Weygolt and Paulus; see also e.g. Dunlop 1997, Dunlop and Lamsdell 2017, Aria and Caron 2019), and unless you’re willing to amend

the taxonomy (which I would *not* recommend doing because there is no need to), you have to remove this statement.

- As stated above, our research does not suggest or support the unique synapomorphy of Euchelicerata. We kept the statement, however we rephrase the sentence and acknowledge the hypotheses supported in Dunlop 1997, Dunlop and Lamsdell 2017, Aria and Caron 2019.

253 – 254 – Dibasterium and Offacolus were considered euchelicerates before, and were already forming a clade basal to “Prosomapoda.” And obviously Euchelicerata was already monophyletic. You should focus on the novelties.

We feel that keeping this short text is justified and ensures a clear understanding of the implications of our research, so we left this text in with only slight modification. As clarified earlier in our comments , our results show that *Setapedites*, even lacking the book gills opercula and being part of Offacoliidae, is part of Euchelicerata according to our phylogenetic analyses.

255 – As explained above, you cannot be certain.

Amended. Added new specimen.

257 – Again, refer to the actual diagnosis of Euchelicerata.

See answer to comments above, also there is not an actual diagnosis only proposed once.

260 – A number of these characters are actually shared with Sanctacaris. Why is it absent?

It does not have chelicera, which in our argumentation is defining euchelicerates.

269 – To be clear, you construe that the megacheiran-like exopods of the seventh pair of cephalic appendages in Habelia are homologous to the seventh pair of appendages in “setopods”, implying the reduction of the endopod.

Correct, a secondary loss of the endopods of the seventh appendages in Offacoliidae is implied in this view.

275 – I think it’s fair to cite Aria and Caron 2019 Nature here.

-Amended. Citation to this reference has been added.

285 – Again, see Mollisonia, specifically ROMIP 65262. The taxon has the same “brush” terminations in the prosoma. The absence of the feature in habeliids just has to do with the fact that the exopods are very slender (like exopod #6 in Offacolus), with small tips, but a bunch of setae is present too.

-Amended. We checked the proposed material and considering the generally excellent preservational state of the *Mollisonia* material we consider the presence of this feature questionable and the same is true for *Habelia*, however, we added an acknowledgement of this interpretation in the text.

290 – You identify what possibly is the basipod of an appendage (as was done with Offacolus and Dibasterium already), but nowhere I see that you can demonstrate where exopods attach. Given the resolution of your material compared to the Burgess Shale or the Herefordshire, I think this will be difficult.

-In order to show the basipod more clearly, we have inserted a new 3D rendering of synchrotron μ CT data to show it.

291 – Proximalmost appendage morphology in Habeliids as well as Mollisonia is no less clear than in Offacolus and Dibasterium: in all cases, the semi-independence of the exopod is based on the absence of evidence for the point of attachment and the observation that they do not seem constrained by the

movement of the endopods. Which means that this condition started off with Panchelicerata. Thus the polarization is quite clear. Did this lead to the loss of prosomal exopods? Yes, most likely – this was discussed before (Aria and Caron 2017, 2019, Aria 2021 PaleorXiv).

-Amended. The uncertainty about *Habelia* is now omitted from the paragraph. The suggested evolutionary scenario and references to Aria & Caron 2017 and 2019 were already in the manuscript text at this point.

294 – What about exite instead of exopod?

Paddle-like appendages in both *Dibasterium* and *Offacolus* are considered to be the exopodal in origin. We do agree with this interpretation and follow it.

306 – Mmmm... but 1) nothing here is new, because this scenario was discussed in the context of *Mollisonia*, and 2) the morphology of opisthosomal limbs in your fossil is unclear beyond a lobate aspect (that could very well be opercula!).

-We acknowledge this scenario to have been discussed in Aria & Caron 2019 adding the citation into the text. We answered the doubts regarding the interpretation of the opisthosomal appendages in other comments above and with new figures and specimens (see S1 G and H and new figure 3)

307 – Whether the name should define the group of taxa or the reverse is always a tautological argument. If you say “Should Euchelicerata represent the least inclusive clade containing XXX”, then I understand, but this is explicitly at odds with the current formal diagnosis. I do see the issue that the absence of opercula in your taxon could render Euchelicerata polyphyletic, but 1) like I wrote above, the presence of opercula remains entirely possible, and 2) this is dependent on the topology and the robustness of the results. As a matter of fact, if you support the monophyly of “setopods” and its polytomy, the presumed absence of opercula in your morphotype would parsimoniously optimize as a reversal (ACCTRAN or DELTRAN), contradicting your statement. Why don’t you optimize important character state changes on your topology? This would make it very clear.

-Amended or answered above. The phylogenetic analyses we have performed, even including *Mollisonia* and using different methodology as suggested by the reviewer support Offacolidae as a monophyletic group at the base of Euchelicerata (sensus Weygoldt & Paulus, 1979). As such, the absence of opercula in *Setapedites* is considered a retained feature. We also add a sentence in the MS exploring the possibility of it being a reversal, hypothesis that we do not support but we cannot exclude.

Figures: Some of your Supp. figure panels are much more interesting to put as main figure panels, e.g. 2C or E (but not 2F) (that synchrotron effort should have more visibility!), 3A-B (for the chelicerae and putative bipartite epistome-labrum), a larger version of 3J. In your main figures 1 and 2, many panels are redundant or don’t show much, so they can easily be replaced.

-Amended. Thank you for the suggestion. We have added the requested images from the supplementary files into the main manuscript Figures 2 and 3. 3D renderings and tomographs from the synchrotron data have now been included in a figure in the main manuscript, Figure 4.

Figure 1. L panel should be K.

-This has been corrected.

Figure 2. In E, please correct tone – this is much too yellow and hampers visibility of features.

-Colour tone has been balanced in this figure.

Figure 2C. If the labrum is the posterior lobe of the structure you divide in 2, what's the anterior portion? The epistome? This is a bit of a rhetoric question, as you've probably reached the limit of informative resolution here. I think you're better off calling the entire area putative epistome-labrum (if you want to keep with chelicerate terminology, or any other term as long as it's justified).

- A detailed discussion of the labrum region has been added to the supplementary information. Possible interpretations for both structures mentioned in the reviewers comment are now discussed.

Supplementary Figure 3. Drawings G and O are overinterpretations. You draw all boundaries as if those were genuine, but it is evident that the structures are severely altered and that you cannot make the case for these extra long spines, etc.

-In order to show the presence of these anatomical structures, we have provided better-quality photos that better support our interpretive drawings.

Supplementary Figure 4. Put the PPs everywhere, not just for the Chelicerata clade.

-Amended.

Supp. discussion – Prosoma ground pattern: Habeliidans and mollisoniids, as well as some other unpublished Burgess Shale taxa, have seven pairs of appendages under the cephalic shield. Should these taxa form a clade, this could be indeed autapomorphic. But it is also not unfathomable to think that this tergite was secondarily released—this is commonplace among crustacean cephalothoraxes! I don't think there is a need for you to oppose the evidence so strongly, especially when it's otherwise very clear in many specimens and several taxa.

-We amended the discussion to tone down our opposition to this hypothesis.

I hope these comments were fair and constructive. I will be glad to see a revised version of this work and, if satisfying, recommend it for publication.

We thank Cédric for these helpful comments, which we believe have greatly improved the manuscript and clarified our arguments. We addressed all comments as fairly and extensively as possible, while also still maintaining some of our original line of thinking, in the few places where we have to respectfully agree to disagree with the reviewer on one or two points. We find our manuscript now presents a better balance between our point of view and that of Cédric. We hope he finds our revised manuscript to be satisfactory.

Cédric Aria

Reviewer #2 (Remarks to the Author):

This is a nice manuscript describing a very common synziphosurine from the Fezouata biota. The record is important for a number of reasons in terms of both its age and the quality of preservation.

The manuscript is well written and well presented. I am not a specialist on chelicerates so I decided to simply read the manuscript and see if the presentation was sound and whether I could identify the characters in question.

We thank Jakob for his supportive comments, and we appreciate hearing his suggestions on the overall presentation and argumentation in the manuscript. We believe the manuscript is even stronger after having incorporated his suggested changes.

The specimens are really small. Hence the appendages are faint infill of iron oxides and clays after pyrite. I bet some referees may say they can't see the podomere boundaries and while the images are not always the best ones I can see them and believe the authors' interpretation. So, in case you get some obstinate referees saying that they can't see a thing, tell them that its because they are blind or at least not in the possession of trained eyes to look at exceptional fossils and hence should abstain from commenting on such issues again.

Many thanks for our words of support here. Fortunately, we did not receive major opposition to most of our anatomical features. Some comments from other reviewers made suggestions for specific improvements to the photographs, and so we have hopefully generally increased the quality of our photographs and figures. A few anatomical features were challenged by other reviewers (e.g. bifurcate telson tip) in response to which we added some new photos or new specimens. New 3D rendering from synchrotron data have also been added.

I can see the biramous prosomal appendages and the chelicerae. I am struggling a bit more with the opisthosomal appendages and their morphology, but agree that there is some structures that clearly derives from exposure of structures below the tergal skeleton and which is appendicular in nature. The supplemental figures are more helpful.

We are happy that the supplemental figures clarify our interpretation of the opisthosomal appendages. In this regard, we add a new specimen to clarify even more this anatomical feature(Figure S1 G and H).

Too bad these couldn't have been revealed with the synchrotron. I surely would have taken 20-40 individuals to the machine after having cut them down to size and scanned the hell out of them. Then I would have manually segmented them out with a nice bottle of Swizz white wine by my side rather than thresholding them which often miss more faintly preserved features. Surely a lot of cool stuff would be revealed that way and some specimens could reveal more appendicular details, but given the approaches done here, this manuscript is sufficient for the description to be useful for other researchers, so I will not linger on this thought or suggestion for longer.

-Amended. In order to better illustrate the anatomical features identified by the reviewer as unclear, and following his suggestion, we have deeply reworked the segmentation of our synchrotron μ CT scan, and have also obtained additional scans for other specimens. Among the new scans, which exhibit a strong heterogeneity in terms of the amount of details visible for the cephalic appendages, we have manually segmented a second specimen, preserved in dorso-lateral view. We have dedicated a whole new figure (Figure 4) to this aspect, including several 3D renderings and tomographs from these datasets. This revealed the opisthosomal appendages and other structures in better detail (see also our replies to the comments of the Cedric Aria).

I notice that the material is supposedly all based on collections made in 2015 and 2016 and kept in both Lausanne and Yale. Now, I believe a lot of the material at Yale was collected in 2009 and up to 2011/2012.

I was involved in collecting some of this material with PVR and I remember specimens of similar quality as the ones shown here and I even have a couple of specimens in Bristol preserving nice limbs. I wonder why this material was not included or mentioned?

We included all material that was available to us based on our study of the Lausanne and Yale specimens. This includes all the YPM material collected by Ou Said Ben Moula, his family, Peter Van Roy and others prior to 2015 (that is accessioned at Yale) and we have amended the text to better reflect that the entire collection in Yale was examined. We were not aware of material accessioned in collections at Bristol. Generally speaking, the major focus of this work was on the Lausanne collection, and while we did make sure to include all relevant and important specimens from the Yale collection, we did preferentially choose Lausanne specimens to illustrate in the manuscript when material of equivalent quality was available from both collections.

Minor additional comments.

Figure 2 labels are confused. I was looking for the opisthosoma appendage panels (Fig. 2E,F), but the labels indicating those panels are missing and then the subsequent panel labels follows a much higher set of alphabetical letters. I guess some editing between journals led to this omission.

-We have corrected the labelling in this, and double checked labels in all figures. Thank you for pointing out this error.

I don't see any reason to hold this manuscript up from what I can gather. Unless there is an issue with the taxonomic work, which someone like David Legg, Greg Edgecombe, Jason Dunlop, James Lamsdell or such noticed I think it is fine to publish with a few tidbits to be clarified. I don't need to see the manuscript again.

Thank you again for your comments and for your support of this manuscript.

Best

Jakob Vinther

Reviewer #3 (Remarks to the Author):

What are the noteworthy results?

This is the first formal description of this common element of the Fezouata biota, a very important Ordovician Fossil-Lagerstätte. Comparison of other early Palaeozoic euecheliates with appendages revealed that the new genus provides important information concerning the evolution of euecheliate appendages.

Will the work be of significance to the field and related fields? How does it compare to the established literature? If the work is not original, please provide relevant references.

The work is significant because it adds to our knowledge of early euecheliates.

Does the work support the conclusions and claims, or is additional evidence needed?

In general, but the new genus has only been compared, at least in the text (the supplementary material is simply NEXUS files), to other synziphosurines with appendages.

- A supplementary file had previously been provided showing additional figures and with additional text, some of which extended the discussion to taxa before the other synziphosurines. We regret to hear that the reviewer did not have access to the supplementary file and we hope that it will be provided to the reviewer, if necessary, in its revised version. Based on comments from other reviewers, we have revised and in some places expanded our discussion of the wider significance of this taxon.

Are there any flaws in the data analysis, interpretation and conclusions? Do these prohibit publication or require revision?

The main problem with this work is that, at first sight, the new genus looks remarkably like other synziphosurines, such as *Pasternakevia* Selden & Drygant, 1987 (poorly redescribed by Krzeminski et al. 2010) and aglaspidids such as *Tremaglaspis* Rushton & Fortey (see, for example, Lerosey-Aubril et al. 2013), and others. Most of these do not show appendages, but this is no reason to exclude them from the analysis. How do we know they are unrelated to the new genus? I would like to see discussion of these genera, which greatly resemble the new form, at least in dorsal morphology.

-We welcome the reviewer's suggestion here to broaden our discussion of other synziphosurines and their similar dorsal morphology. A new section has been added to the Supplementary Information directly addressing the taxa listed above and comparing them to our new *Fezouata* taxon.

Is the methodology sound? Does the work meet the expected standards in your field?

See comments above re comparison with other synziphosurines and aglaspidids.

The requested comparisons have been added to the supplementary text.

Is there enough detail provided in the methods for the work to be reproduced?

Yes.

References

- Krzemiński, W., Krzemińska, E., and Wojciechowski, D. 2010. Silurian synziphosurine horseshoe crab *Pasternakevia* revisited. *Acta Palaeontologica Polonica* 55 (1): 133–139.
- Lerosey-Aubril, R., et al. 2013. "Occurrence of the Ordovician-type aglaspidid *Tremaglaspis* in the Cambrian Weeks Formation (Utah, USA)." *Geological Magazine* 150(05): 945-951.
- Rushton, A. W. A. and R. A. Fortey 2003. "A new aglaspidid arthropod from the Lower Ordovician of Wales." *Palaeontology* 46(5): 1031-1038.
- Selden, P. and Drygant, D.M. 1987. A new Silurian xiphosuran from Podolia, Ukraine, USSR. *Palaeontology* 30: 537–542.

We thank the reviewers for their comments and for providing the necessary references, which we are sure to include in the revised version of the manuscript and its supplementary material.

Reviewers' Comments:

Reviewer #1:

Remarks to the Author:

The authors have been respectful with my comments and have generally taken care to address them, although some critical points have been left wanting. There remains, in my view, substantial work to do to make the study sound and tight. In particular, looking now more attentively into the fossils, critically, some interpretations have to be reconsidered, and most likely revised. Unfortunately, this also means the cladistic analyses will have to be rerun. Camera lucida drawings also need to be improved, because as it stands, they are not accurate enough.

As preamble, and especially considering the preservation of chelicerae documented in this very paper, I regret that the authors doubt the well-preserved chelicerae of *Mollisonia* but not the "evidence" for nervous tissues in specimens shown in Ortega-Hernández et al. 2022, *Nat Comm*, which is thoroughly non-existent. As the authors are aware, circum-intestinal carbon films with branching ventral tonguelets are ubiquitous among BST arthropods and have already been documented extensively (since before Whittington...)—the specimens in question fall within that taphonomic range, not to mention other inconsistencies and misinterpretations in this publication. Nevertheless, it is the current authors' prerogative to follow these views, and it is by all means presented fairly. In places, however, the authors need to be more explicit about the argumentative ground for their disagreement, in the stead of broad dismissal – see below.

> That might be seen as peripheral, but to me it is important. It does not serve you to inflate your title. If we did not know *Offacolus* and *Dibasterium* already, perhaps you would be in your right to say that your fossil was "revealing" early euchelicerate diversity. Besides, you do not provide more detailed insight than the Herefordshire taxa. Expands, illustrates, even "evinces", all of those are fine. Again, this does not change anything to the importance of your findings, and really improves the rigor of your output.

> I would not discard the *Arachnopulmonata* result. This is a very current (and sensitive!) debate, and this issue will pop in the mind of many interested readers (especially neontologists). It is not much work to add another tree to your supplementary and strengthens your paper by showing that your results are robust to the position of *Scorpiones*.

> An exopod with brush-like termination is shown in Aria and Caron 2019, *Nature*, Fig. 2g, ED Fig. 3f. This is certainly as well preserved as in your specimens. What is there to be skeptical about? What is your alternative interpretation?

In certain cases of high variability of traits, it is indeed warranted to diagnose by character state combinations, but clearly this is not the case of exopodial setal brushes, present in habeliidans and very much arguably so in mollisoniids. Considering the habeliidan setal brush as analogous based on size difference (?) is simply untenable and even ad hoc justification. It is also the case for biramy and these exopods being stenopodous—these states are plesiomorphic, clearly present in the common euchelicerate ancestor (emphasizing the euchelicerate assignment does not change that fact).

Next to this, the six-podomereous exopod character needs revision. Briggs et al. counted 5 in *Dibasterium*, the fifth being a separately branching podomere, not counting setal tip, but exact podomere boundaries (especially pod. 3) were not entirely unambiguous. Interestingly, *Offacolus*' prosomal exopods purportedly (but convincingly) had six podomeres, without setal termination, but the designated pod. 5 is identical morphoanatomically to pod. 4 of *Dibasterium*, being the one producing the branching out distal podomere. As a result, both indeed likely had 6-podomereous exopods *excluding termination*—however, in Aria and Caron 2017 *BMCEB*, limb terminations (claws or setal brushes) were included in the podomere count as has been done traditionally. There is therefore ground for heptapodomereous exopods being plesiomorphic too.

Then the problem, as I explain below, is that you interpret exopods in Fig. 2G, H when they seem in fact to be endopods.

Traits that hold for Offacolidae: elongate chelicera, reduced prosomal gnathobasipods, seventh prosomal appendage pair reduced to setose paddle exopod, bifurcate telson tip.

For Setapedites: See my comments below. You make a case for the raised tergum, and segmental anatomy as well as the shape of cephalon (closer a priori to *Dibasterium*) should be emphasized here. The shape of the 11th opisthosomal tergite indeed also counts in. It is unclear to me how you differentiate the short triangular pleurae from said sub-axial nodes.

> The interpretative drawings of Figure 1 are inaccurate—See my annotated pictures attached. There are three regular metasomal segments followed by a larger plate, as well as possibly intercalary elements between this plate and the telson (likely including an additional segment and a telson head). Of the mesosomal segments, 6 are well exposed, while an anterior one is mostly covered by the occipital margin of the head shield. However, MGL102872 also shows an additional element of what seems to be a tergo-pleura anterior to that segment (thick arrow). This very sensibly can make the link between the habeliidan body plan and the offacolid one, insofar as you seem to have 12 trunk segments preceded by a separate segment integrated into the cephalon, but apparently still expressed here. This is in fact a solid and interesting find! On my image, db is double - it is nicely preserved, you should mark it.

> Overall, morphoanatomy seems close to *Dibasterium*, including the ventral side. It appears indeed that all mesosomal segments bear appendages, so 7 pairs. Here again, I believe your interpretative drawings are inaccurate, or at least incomplete (see annotated image). Given the disposition of segments, it seems very likely that the 7th appendage pair, borne by a very reduced segment, is tucked mesially, as in *Dibasterium*. Alternatively, you could be right about this first large pair being the 7th prosomal pair, implying that the other pairs are attached a bit more anteriorly, thus leaving you with an anatomy comparable to *Dibasterium* (the last mesosomal segment lacking appendages). It also appears to me that the triangular shaped formed by the juxtaposition of reduced endopods mesially (as in limulids) is visible—you drew one of them, but not the others...

> In MGL102634 (Fig. G, H), I think that what you see are the endopods, not the exopods. The podomeres are thick and the distalmost podomere looks like a stout claw, bordered by setae on a small platform. This is similar to what is known in habeliidans. Most importantly, these appendages are attached very mesially, close to the central axis—as is expected of endopods seen from ventral view. In Fig. 3A, B, these indentations would correspond to the tougher masticatory margins of the basipods, which tend to be more three-dimensionally preserved in compression fossils. I reckon you're interpreting these as folded endopods, based on *Offacolus* and perhaps extant horseshoe crabs, but consider the differential size—to me, that doesn't match. This is also consistent with what you get from X-ray tomography, in Fig. 4B-E: These sclerotized wedges are the gnathobasipods. In Fig. 2I, J, you draw exopods and endopods sharing a basis. Given the evidence, this seems a conjecture, especially in the context of odd relationship between these rami. Your CT data, however, are better in that regard; I don't see exopods attaching to a gnathobasipod, but it is fair to say that the proximalmost part of the exopods is very close to that of the endopods.

I'm attaching a collage to compare the limbs in MGL102634 with the exopod of *Offacolus* and the endopods of *Habelia*. Please carefully consider.

> I am sorry to say, but calling the chelicerae in *Mollisonia* "poorly-preserved" while taking for granted all the fossil evidence provided in this very paper is unfair to me. As a matter of fact, the outline of chelicerae, including chelate elements in dorsal view, is better preserved in *Mollisonia* than in the fossil described here. Budd's arguments were responded to in Aria 2022, Biological Reviews. If you still think

these elements could be something like non-appendicular protocerebral protrusions, say so, but at least give an argument.

Regarding the pygidium, I am not sure what the argument is exactly, but *Mollisonia*'s pygidium is four-segmented (not three-segmented as written here), as shown by the number of appendage pairs and structure (Aria and Caron 2019, *Nature*, Figs. 1d,e, 2i, ED Fig. 4). It is exactly the same as in any other arthropod with a pygidium...

> Back to book gills and Euchelicerata... First of all, Euchelicerata is taxonomically diagnosed by the presence of opercula, so you cannot write that your fossil would demonstrate this is not the case—this makes no sense. This is not Aria and Caron (2019) who suggested it, it was formally defined by Weygoldt and Paulus (1979: *Untersuchungen zur Morphologie, Taxonomie und Phylogenie der Chelicerata*. *Z. Zool. Syst. Evol-forsch.* 17 (85-115), 177-200.). This is a rule of systematics that you have to follow.

Now, if you were to find that gill opercula arose convergently, what you could say is that Euchelicerata, as currently diagnosed, is polyphyletic (I already specified that in my first review...). You may also emend the diagnosis to add or replace diagnostic traits. However, and again, you need to be more circumspect here, and admit that your evidence is not good enough to be unambiguous. In MGL107741, those well-defined paddle-shaped elements appressed onto the ventral side could very well be opercula—which is simply defined as a semi-rigid lobate sheet covering lamellae. What you identify as a setose margin is quite tenuous, and would not in fact rule out an operculum in itself.

Nevertheless, and again I am repeating what I wrote in my first review, even if you coded with certainty the absence of operculum in your species, based on parsimony this optimizes as homoplasy! I tried to show that using your Lamsdell dataset and tree result... only to find out that the operculum is not even present in this matrix. How could you test anything related to the evolution of the operculum if you don't even code it in your dataset?

So I used the other dataset, from Aria and Caron 2019, and looked at the optimization of opercula for both parsimony and likelihood on the Bayesian result. See attached figure. Likelihood optimization is extremely strict and does not achieve more than ambiguity, but a parsimony-based optimization clearly places the presence of opercula at the Euchelicerata node as retrieved previously in Aria and Caron 2017, 2019, with the ambiguity in *Setapedites* being a reversal.

Similar story for the peculiar exopod attachment condition. This most parsimoniously optimizes as ancestral for panchelicerates—sea spiders notwithstanding—despite you recoding uncertainty in *Mollisonia*. This evolutionary pattern was put forward as the illustration of the possible mechanism for the loss of the outer branch in the euchelicerate prosoma in Aria and Caron 2017, 2019, an idea that you misleadingly claim to be new here.

Additionally, if *Mollisonia*'s trunk appendages are not multi-lamellate structures akin to book gills, what are they? I am defender of a good science based on open refutation, but refutation should be explicit, fact-based.

> There are issues with the use of "stem" and "crown" terminology throughout, which I think has more to do in fact with the taxonomic issue of what Euchelicerata means as explained above. First, in your Fig. 6, Synziphosurines are not part of the crown, since they left no extant descendants. Second, as I demonstrate above, likelihood and parsimony optimization still places Euchelicerata at the node joining up offacolids and prosomapods. Third, considering that you're leaving Chelicerata out of the picture, you need to include early panchelicerates *Mollisonia* and *Habelia* into the stem of euchelicerates.

In summary:

- You need to be more explicit about your rejection of the evidence in *Mollisonia*.

- The evidence is not clear enough to ascertain the absence of opercula. This should be left more open.
- If you actually optimize your character on your tree (which I suggested you to do in the first round but you did not), you find that opercula are present in the common ancestor of Euchelicerata sensus Wygoldt and Paulus and as retrieved by Aria and Caron 2019. This contradicts your statements.
- Same for the prosomal exopod condition: Most parsimoniously this is present in the common ancestor of panchelicerates, again as stated in Aria and Caron 2019.
- Improve your interpretative drawings and reconsider segmental anatomy, as well as prosomal gnathobasipod and endopod identification.

I refrained from examining the MS in finer details – language, references, etc. The authors must first address these remaining salient problems, some of which being carried over from the first round of reviews. Should they do so, I expect that only one additional round will be necessary for said details. I want to make clear that I am not trying to impose my views, and I would also rather not spend a week on a review, but I am trying my best to show the authors that the shortcomings I point out are not simply a matter of opinion—as they should not be, on either side.

Other specific comments:

I.38: Offacolidae

I.51: The statement that it is “obscure” washes away the work on both the Herefordshire and the Burgess Shale. It remains poorly documented, or something equivalent.

I.51: Regarding the definition of Euchelicerata here and hereafter, see my comment and correct.

I.56: You refer to phylogenetic results that are 10 years old—and in the case of Haug et al. 2012, this is not even a cladistic study. Why not acknowledge that these hypotheses have not been recovered again? What recent phylogenetic study has disputed that Habeliida is the closest sister group to Euchelicerata? (Mollisonia notwithstanding). Even if you reject the evidence provided by Mollisonia, Panchelicerata as defined by Aria and Caron (2019) remains valid, yet there is no mention of it here. This grouping was recovered in most recent cladistic work by Zeng et al., Moysiuk and Caron, etc.

I.82: The anal pouch is completely speculative. Without redundancy of evidence, I don’t see how this can be differentiated from an artefact. Consider it putative if you want, but just in the figure caption; no need to dwell on that.

I.83: “Setapedites abundantis resolves the position of several problematic stem-euchelicerate taxa”: What do you mean? Setapedites has no phylogenetic impact on the placement of early panchelicerates or other offacolids based on recent phylogenetic results.

Sys. Pal.: See main comments.

I.285: This makes no sense. It’s not because frogs have anterior members homologous with birds that it would play a role in supporting their bird affinity! Also, the statement above is contradictory with what you say elsewhere about gill opercula not being apomorphic of euchelicerates.

I.290: Well, no, they are shared with Sanctacaris, and the enlarged gnathobasipods also shared with Mollisonia. Or do you also contest the presence of large gnathobasipods in Mollisonia?

I.301: Not sure what kind of bilobation you mean in Habelia.

I.327: No, it does not; quite the opposite in fact. If you posit the clear attachment to a basipod in Setapedites, which I contest, it is another autapomorphy, and a reversal, (like the putative absence of opercula) amid the condition shared by offacolids, habeliids and mollisoniids.

I.337: The seventh cephalic pair of early panchelicerates has already been argued to be ancestral to these features in Aria and Caron 2017, 2019. This was one of the main findings in these papers...

Phylogenetic figures: Are you sure you are displaying Maximum clade credibility consensus? The default output from MrBayes is extended majority rule, and the topology coming out of my basic run is the same as yours.

Additionally, happening to be, I think, the “someone like David Legg, Greg Edgecombe, Jason Dunlop, James Lamsdell or such”, I was asked to check whether the authors had fully addressed the comments of reviewer #2, Jakob Vinther. I can’t say about the bottle of “Swizz” white wine, but it seems the

authors took care of providing a better rendition of the raw CT data, as well as adding another specimen—which is, I think, what Dr. Vinther suggested. Clarification was also provided regarding the material used, and labelling in Figure 2 was corrected.

Cédric Aria

A

1
2
3
4
5
6

?? ??

MS1

MS2

MS3

MS4

MS5

MS6

MS7

MT1

MT2

MT3

MT4

MT5?

C & D

Opercula: Parsimony

Opercula: Mk model

Detach. exop.: Parsimony

Detach. exop.: Mk model

Reviewer #3:
None

Point-by-point responses to the Referees' comments and suggestions.

We present the referees' comments in normal typeface and **our responses in green bold**.

Reviewer #1 (Remarks to the Author):

The authors have been respectful with my comments and have generally taken care to address them, although some critical points have been left wanting. There remains, in my view, substantial work to do to make the study sound and tight. In particular, looking now more attentively into the fossils, critically, some interpretations have to be reconsidered, and most likely revised. Unfortunately, this also means the cladistic analyses will have to be rerun. Camera lucida drawings also need to be improved, because as it stands, they are not accurate enough.

We are happy that our edited version generally pleased the reviewer. We have considered all his new comments and modified our text and figures accordingly everywhere we felt the demands were legitimate and not the reflection of his own opinion which we (and the majority of the community) do not endorse. Regarding this particular point of the cladistic analyses, the new requested analyses are no longer needed after we removed our statements about the presence/absence of opercula and gills anatomy following other demands by the reviewer (see below). However, we did rerun the analyses after recoding *Mollisonia* (detailed later in our response).

As preamble, and especially considering the preservation of chelicerae documented in this very paper, I regret that the authors doubt the well-preserved chelicerae of *Mollisonia* but not the “evidence” for nervous tissues in specimens shown in Ortega-Hernández et al. 2022, Nat Comm, which is thoroughly non-existent. As the authors are aware, circum-intestinal carbon films with branching ventral tonguelets are ubiquitous among BST arthropods and have already been documented extensively (since before Whittington...)—the specimens in question fall within that taphonomic range, not to mention other inconsistencies and misinterpretations in this publication. Nevertheless, it is the current authors' prerogative to follow these views, and it is by all means presented fairly. In places, however, the authors need to be more explicit about the argumentative ground for their disagreement, in the stead of broad dismissal – see below.

It should be noted here that:

- 1) we added these sentences (lines 260–266) about the affinities of *Mollisonia* in our revised manuscript because (as stated in our cover letter) we believe that it was important to incorporate a mention of (and cite) the personal view of the reviewer on the matter even though it does not reflect the majority view of the community, including ourselves;**
- 2) as shown by our revised cladistic analyses, the inclusion (or not) and coding of *Mollisonia* have no impact on the conclusions of our work.**

That said, we confirm that we are not convinced by the presence of chelicera in *Mollisonia*, but we acknowledge that our wording may let the reviewer think that we fully support Ortega-Hernández et al. (2022)'s interpretation, which was not our intention. Nonetheless, the re-evaluation by Ortega-Hernández et al. (2022) is only one potential opposite view for the interpretation of *Mollisonia* as a chelicerate. Indeed, we also refer to (and cite) a paper by Lerosey-Aubril et al. (2020) that highlights differences in the homology of the pygidium. Furthermore, Budd (2021) also cast doubts about Aria & Caron (2019)'s interpretation: “it is difficult to understand how the putative chelicerae are meant to have functioned, as they seem too far away from

the mouth to be used for manipulating food; and definitive evidence for the structures actually being chelate or having podomeres is lacking. The possibility of these structures being pre-antennal outgrowths of the sort discussed by Ortega-Hernández and Budd (2016), is not considered. In addition, a central rounded structure between the two supposed chelicerae is dismissed without evidence as being an artefact, even though it is symmetrically placed in the organism and indeed appears to be present in more than one specimen (i.e. MCZ, 1811 and ROMIP 62978). Thus, the phylogenetic position of *Mollisonia* seems presently unclear". We did not, however, refer to this opinion by Budd (2021) in our revised manuscript, and have now added a mention in our new revised manuscript (see below).

All three papers mentioned above cannot be considered irrelevant, and the ambiguous affinities of *Mollisonia* remain a fact within the scientific community. Considering that there are not enough consistent arguments to fully support either view, we feel (as stated in our cover letter) that both should be presented in our work. We also included in our new revised manuscript a mention to a highly provocative and controversial paper published by the reviewer after the submission of his review (Aria et al. 2023, BioEssays), in which they provide a contrarian view on the fossilization of nervous systems.

Considering all these points, we have modified this section (I260–266) as follows:

"Among Cambrian euarthropods, *Mollisonia plenovenatrix* has been described as possessing a pair of short chelicerae and proto-book gills composed of overlapping exopod flaps, and retrieved as a basal chelicerate (Aria & Caron 2019). Counter arguments include the poor preservation of these features and functional considerations (Budd 2021), the organization of the central nervous system in *Mollisonia symmetrica* (Ortega-Hernandez et al. 2022) (but see Aria et al. 2023), as well as the origins and development of the last three segments complex (pygidium) in the genus *Thelxiope* and *Mollisonia* (Lerosey-Aubril et al. 2020). In any case, the inclusion (or not) and coding of *Mollisonia* have no impact on the conclusions of the present work."

> That might be seen as peripheral, but to me it is important. It does not serve you to inflate your title. If we did not know *Offacolus* and *Dibasterium* already, perhaps you would be in your right to say that your fossil was "revealing" early euchelicerate diversity. Besides, you do not provide more detailed insight than the Herefordshire taxa. Expands, illustrates, even "evinces", all of those are fine. Again, this does not change anything to the importance of your findings, and really improves the rigor of your output.

We have modified our title accordingly: *The oldest-known synziphosurine reveals early euchelicerate diversity and evolution*

> I would not discard the Arachnoplumonata result. This is a very current (and sensitive!) debate, and this issue will pop in the mind of many interested readers (especially neontologists). It is not much work to add another tree to your supplementary and strengthens your paper by showing that your results are robust to the position of Scorpiones.

We have added the requested tree as a new Extended Data figure.

> An exopod with brush-like termination is shown in Aria and Caron 2019, Nature, Fig. 2g, ED Fig. 3f. This is certainly as well preserved as in your specimens. What is there to be skeptical about? What is your alternative interpretation?

Although we indeed find it difficult to determine whether the feature in Aria & Caron 2019 represents a subchelate, spinose, or trilobite-like termination, we have now included a sentence in the main text mentioning their presence and using the description of Aria & Caron 2019, as follows “*Mollisonia* also shows a single exopod in one specimen with a termination made of setae ⁶, but their precise arrangement are unknown. Possessing exopodial setal brushes may be homologous amongst these taxa, but we interpret the precise morphology of stenopodous exopods 2–5 ending in a brush-like group of long and radially arranged setae to be an autapomorphy of Offacolidae ...”.

We have additionally rerun both matrices using the original coding of Aria & Caron 2019 for *Mollisonia*, and we include these results in the supplementary information as Supplementary Fig. 10 and 11. So now we show his view, as well as our own view. Overall, the trees change very little when the original codings for *Mollisonia* from Aria & Caron are used. Therefore, the text in Supplementary Information describing the character coding change for *Mollisonia* has been changed to indicate we use their original coding, as follows:

“Character 123- Stenopodous exopod type coded as both ? and 1 (respectively Supplementary data 1 and 2)

We ran two Bayesian analyses, one with this character state coded as ?, resulting in the trees of Supplementary Fig. 5-8 and another where this state is kept as 1 as originally coded in Aria & Caron (2019) and resulting in Supplementary Fig. 10 and 11. This is because, in our opinion, this structure (described from a single exopod in one specimen in Aria & Caron 2019) is difficult to interpret in detail but seems to have a different structure to that seen in *Setapedites abundatis*.”

“Character 69- Exopod on second post-antennular limb coded as both 2 and 3 (respectively Supplementary data 4 and 5).

We ran two Bayesian analyses, one with character state 3 (reduced state of prosomal exopod), resulting in the trees of Fig. 6 and Supplementary Fig. 6, 7 and 8, and another where this state is changed to 2 (pediform exopod), resulting in the trees of Supplementary Fig. 11. These two different codings reflect different interpretations of *Mollisonia* cephalic exopods in comparison with offacolids. The two different codings however do not change the result of the analyses for the target group.”

In certain cases of high variability of traits, it is indeed warranted to diagnose by character state combinations, but clearly this is not the case of exopodial setal brushes, present in habeliidans and very much arguably so in mollisoniids. Considering the habeliidan setal brush as analogous based on size difference (?) is simply untenable and even ad hoc justification.

Regarding the setae condition in *Mollisonia*, please see our reply to the previous comment.

Regarding the condition in habeliidans, we thank the reviewer for his comment that helped us strengthen our manuscript. Setae in *Habelia* are definitely much shorter (shorter than the podomeres) than those in Offacolidae (longer than the podomeres). Sutton et al. (2002) described them for Offacolus as “a brush-like splay of long setae”, a terminology that we are happy to follow in our revised text. But, more importantly, another significant difference between *Habelia* and Offacolidae is that setae are confined to the inner margin of the exopod in *Habelia* whereas they are radially arranged from the inner to the outer margins in Offacolidae. We have made this clearer in our manuscript as follows:

- lines 96–97, “stenopodous exopods of six podomeres bearing a brush-like group of long and radially arranged setae on the distalmost podomere.”
- lines 316–319, “Exopodial setal brushes are also present in *Habelia*¹². However, they are short and confined to the inner margin of the exopod in *Habelia* whereas they are long and radially arranged from the inner to the outer margins in Offacolidae, clearly differentiating these features.”
- lines 321–323, “We interpret the morphology of stenopodous exopods 2–5 ending in a brush-like group of long and radially arranged setae to be an autapomorphy of Offacolidae, a conclusion which is supported by our phylogenetic analyses (Fig. 6, Supplementary Fig. 5–8).”

It is also the case for biramy and these exopods being stenopodous—these states are plesiomorphic, clearly present in the common euchelicerate ancestor (emphasizing the euchelicerate assignment does not change that fact).

We fully concur with the reviewer and also recognize the stenopodous condition as likely plesiomorphic of euchelicerates and shared by habeliidans and offacolids, and possibly *Mollisonia*. The manuscript text states this already in the discussion “Biramous prosomal appendages with specialized exopods (with respect to the ancestral condition for stem euarthropods⁴³), together with a uniramous first pair of appendages (cheliceræ) and uniramous appendages of somite VII made of bipartite paddle-like exopods, can therefore be considered as part of the euchelicerate ground plan.”

Next to this, the six-podomerous exopod character needs revision. Briggs et al. counted 5 in *Dibasterium*, the fifth being a separately branching podomere, not counting setal tip, but exact podomere boundaries (especially pod. 3) were not entirely unambiguous. Interestingly, *Offacolus*' prosomal exopods purportedly (but convincingly) had six podomeres, without setal termination, but the designated pod. 5 is identical morphoanatomically to pod. 4 of *Dibasterium*, being the one producing the branching out distal podomere. As a result, both indeed likely had 6-podomerous exopods *excluding termination*—however, in Aria and Caron 2017 BMCEB, limb terminations (claws or setal brushes) were included in the podomere count as has been done traditionally. There is therefore ground for heptapodomerous exopods being plesiomorphic too.

We concur that the designated pod. 5 in *Offacolus* is identical morphoanatomically to pod. 4 of *Dibasterium*. Looking carefully at the exopods of *Dibasterium* in Briggs et al. (2012), in particular the one depicted in fig. 1M, a 6th podomere could well be present between those labelled 4 and 6, corresponding to pod. 5 in *Offacolus*. Therefore, if following the reviewer's suggestion we count the setal tip as a separate podomere, we find seven podomeres in Offacolidae, whereas Aria & Caron (2017) indicated that exopods in *Habelia* “are represented by five long and slender rami made of seven or more podomeres”. Considering that “seven or more” may be seven, then matching the number in Offacolidae, we deleted our statement l319–321 (“Other than the distinctive brush-like podomeres, the number of podomeres is higher in *Habelia optata* (7 or more) than in Offacolidae.”), which had no impact on our conclusions.

Then the problem, as I explain below, is that you interpret exopods in Fig. 2G, H when they seem in fact to be endopods.

Based on the distinctive morphologies of the endopods and exopods in our fossil (which is visible in dozens of specimens), we are absolutely certain that at least the

main highlighted appendage branches are exopods. Their morphology perfectly matches that described for *Offacolus* exopods. No change was made in response to this comment.

Traits that hold for Offacolidae: elongate chelicera, reduced prosomal gnathobasipods, seventh prosomal appendage pair reduced to setose paddle exopod, bifurcate telson tip.

We agree with these traits. Following the previous answer to previous comments, we preserve the following diagnosis for the group: “Euchelicerates with elongate chelicerae, at least second to fifth prosomal appendage pairs being biramous and with reduced gnathobasipods and composed of stenopodous exopods of six podomeres bearing a brush-like group of setae on the distalmost podomere. Seventh pair of appendages (sixth post-cheliceral appendage pair) uniramous and lobate paddle-like, fringed by lateral spines. Tip of the telson bifurcate. Diagnosis modified from Sutton, et al. 2002.”

For Setapedites: See my comments below. You make a case for the raised tergum, and segmental anatomy as well as the shape of cephalon (closer a priori to Dibasterium) should be emphasized here. The shape of the 11th opisthosomal tergite indeed also counts in. It is unclear to me how you differentiate the short triangular pleurae from said sub-axial nodes.

As illustrated in Fig. 1 E-F, the sub-axial nodes are rounded and located at the junction between the bipartite tergites and the triangular pleurae. We have made the “saxn” label line longer to locate more clearly the sub-axial node in the drawing shown in Fig. 1F.

> The interpretative drawings of Figure 1 are inaccurate—See my annotated pictures attached. There are three regular metasomal segments followed by a larger plate, as well as possibly intercalary elements between this plate and the telson (likely including an additional segment and a telson head). Of the mesosomal segments, 6 are well exposed, while an anterior one is mostly covered by the occipital margin of the head shield. However, MGL102872 also shows an additional element of what seems to be a tergo-pleura anterior to that segment (thick arrow). This very sensibly can make the link between the habeliidan body plan and the offacolid one, insofar as you seem to have 12 trunk segments preceded by a separate segment integrated into the cephalon, but apparently still expressed here. This is in fact a solid and interesting find! On my image, db is doublure - it is nicely preserved, you should mark it.

We thank the reviewer for his enthusiastic re-interpretation of our fossil, which is in fact a dorsal view specimen. The feature that the reviewer identifies as a doublure (db) is instead the groove delimiting the so-called sunken region, which we describe clearly in our text and is a dorsal anatomical feature of offacolids homologous to the ophthalmic ridge of other euchelicerates. It is true nevertheless that this feature should be labelled in our Fig. 1 for clarity, which we have done in the revised version. On the other hand, a doublure (which is a ventral and not a dorsal feature), is indeed present in another ventral-view fossil, and we have added a label on Fig. 2B to highlight it.

Regarding the number of trunk segments (11 in our interpretation, but 12 for the reviewer), we strongly disagree with the interpretation of the reviewer. Our count of the number of trunk segments is based on the observation of over 350 specimens in total. In the annotated pictures sent by the reviewer, we believe the count of 12 results from miscounting some of the segments or counting the anterior part of the telson as

a segment. We added new labels to Fig. 1 and 2 to show this better, but otherwise we do not to change our interpretation of the number of trunk segments.

> Overall, morphoanatomy seems close to *Dibasterium*, including the ventral side. It appears indeed that all mesosomal segments bear appendages, so 7 pairs. Here again, I believe your interpretative drawings are inaccurate, or at least incomplete (see annotated image). Given the disposition of segments, it seems very likely that the 7th appendage pair, borne by a very reduced segment, is tucked mesially, as in *Dibasterium*. Alternatively, you could be right about this first large pair being the 7th prosomal pair, implying that the other pairs are attached a bit more anteriorly, thus leaving you with an anatomy comparable to *Dibasterium* (the last mesosomal segment lacking appendages). It also appears to me that the triangular shaped formed by the juxtaposition of reduced endopods mesially (as in limulids) is visible—you drew one of them, but not the others...

Based on our observation of hundreds of specimens, we favor the second hypothesis proposed by the reviewer, that our fossil is anatomically very similar to *Dibasterium*. Regarding the vestigial endopods, we indeed noticed the presence of what looks like potential endopods, but rather preferred not to over-interpret such small and faint anatomical features, which could also represent remains of the basipods. No change was made associated with this comment.

> In MGL102634 (Fig. G, H), I think that what you see are the endopods, not the exopods. The podomeres are thick and the distalmost podomere looks like a stout claw, bordered by setae on a small platform. This is similar to what is known is habeliidans. Most importantly, these appendages are attached very mesially, close to the central axis—as is expected of endopods seen from ventral view.

In Fig. 3A, B, these indentations would correspond to the tougher masticatory margins of the basipods, which tend to be more three-dimensionally preserved in compression fossils. I reckon you're interpreting these as folded endopods, based on *Offacolus* and perhaps extant horseshoe crabs, but consider the differential size—to me, that doesn't match. This is also consistent with what you get from X-ray tomography, in Fig. 4B-E: These sclerotized wedges are the gnathobasipods. In Fig. 2I, J, you draw exopods and endopods sharing a basis. Given the evidence, this seems a conjecture, especially in the context of odd relationship between these rami. Your CT data, however, are better in that regard; I don't see exopods attaching to a gnathobasipod, but it is fair to say that the proximalmost part of the exopods is very close to that of the endopods. I'm attaching a collage to compare the limbs in MGL102634 with the exopod of *Offacolus* and the endopods of *Habelia*. Please carefully consider.

We do not agree with the suggestion of the reviewer that these are endopods, and we have kept our interpretation of these structures being exopods. We understand that looking only at Fig. 2G-H the appendages could appear as located ventrally and attached very mesially. However, when looking at the whole specimen (shown in Supplementary Fig. 2) there is no doubt that the specimen is actually preserved in dorsal view. We agree with the mesial position of the exopods, but do not see this as an argument for them to be endopods – as both endopods and exopods have their proximalmost parts very close to each other (as the reviewer agrees with based on our CT data). We maintain that these have a basipod to which they attach, and we invite the reviewer to re-examine Figure 2E, F, Supplementary Figure 2G and Figure 4J, where we have additionally labelled the structures we consider to be the basipods.

Regarding the anatomy of the appendages, we find more similarities between the exopod of *Offacolus* and *Setapedites*, than to the endopods of *Habelia*. We draw the reviewers attention to Supplementary Figure 3G,H, which show the complete anatomy of the appendage in a new higher resolution photo provided after the first round of review. The podomeres of *Habelia* endopods are elongated and flare outwards distally in a pronounced manner, as shown in Aria & Caron 2019 figures 2D, G, I, M, whereas the podomeres of both *Offacolus* exopods (Sutton et al 2002, figure 4B) and *Setapedites* exopods are stouter and with a square shape. Also *Setapedites* does not show a stout claw bordered by setae on a small platform, but rather is a series of radially arranged longer spines, as is now described in the new text we added comparing the exopod setal brushes with *Mollisonia* and *Habelia*. The exopod of *Offacolus* also possesses a pair of thicker spines on the inner margin of the preterminal podomere, which is not seen in *Habelia* endopods (which have a platform of small spines) – *Setapedites* also bears this pair of thicker spines on the preterminal podomere (labelled “st” in Supplementary Figure 3H). We made a comparison collage and include it below. For these reasons, we keep our interpretation of these appendages as exopods in *Setapedites*.

> I am sorry to say, but calling the chelicerae in *Mollisonia* “poorly-preserved” while taking for granted all the fossil evidence provided in this very paper is unfair to me. As a matter of fact, the outline of chelicerae, including chelate elements in dorsal view, is better preserved in *Mollisonia* than in the fossil described here. Budd’s arguments were responded to in Aria 2022, Biological Reviews. If you still think these elements could be something like non-appendicular protocerebral protrusions, say so, but at least give an argument.

Our scepticism regarding the state of preservation of the chelicera in *Mollisonia* (lines 260–263) has been removed, and the sentence modified following a previous comment of the reviewer (see above). The new sentence is more factual and devoid of scepticism: “Among Cambrian euarthropods, *Mollisonia plenovenatrix* has been described as possessing a pair of short chelicerae and proto-book gills composed of overlapping exopod flaps, and retrieved as a basal chelicerate (Aria & Caron 2019).” It is followed by a sentence describing the reasons put forward against this interpretation as currently found in the literature : “Counter arguments put forward in the literature against *Mollisonia*

being a basal chelicerate cite the poor preservation of these features and functional considerations (cheliceræ too small and far from the mouth)⁹, the organization of the central nervous system in *Mollisonia symmetrica*⁷ (but see³⁵), as well as the origins and development of the last three segments complex (pygidium) in the genus *Thelxiope* and *Mollisonia*⁸.”

Regarding the evidence for chelicera in our fossil, it must be noted that this is definitely much less controversial than the presence (or not) of chelicera in *Mollisonia*. Indeed, the latter is the first and only description of this critical feature in a Cambrian organism, potentially changing our view of the origin of the chelicerate bauplan. Conversely, the presence of chelicera in a synziphosurine, which is anatomically extremely similar to *Dibasterium* and *Offacolus*, which preserve chelicera in details, is expected. We argue that if extraordinary claims require extraordinary evidence (i.e. for the presence of chelicera in *Mollisonia*) ordinary claims require ordinary evidence, which we provide in Fig. 2C-D and Suppl. Fig. 3A-B (and consistently for two different specimens).

Regarding the pygidium, I am not sure what the argument is exactly, but *Mollisonia*'s pygidium is four-segmented (not three-segmented as written here), as shown by the number of appendage pairs and structure (Aria and Caron 2019, Nature, Figs. 1d,e, 2i, ED Fig. 4). It is exactly the same as in any other arthropod with a pygidium...

We acknowledge that the reviewer might well be correct here, but referred in our manuscript to the work by Lerosey-Aubril et al. (2020) who expressed a different opinion. As this issue is absolutely peripheral to our work and we mention the segmentation condition of the pygidium in *Mollisonia* while referring to Lerosey-Aubril et al. (2020) we made no change to our manuscript.

> Back to book gills and Euchelicerata... First of all, Euchelicerata is taxonomically diagnosed by the presence of opercula, so you cannot write that your fossil would demonstrate this is not the case—this makes no sense. This is not Aria and Caron (2019) who suggested it, it was formally defined by Weygoldt and Paulus (1979: Untersuchungen zur Morphologie, Taxonomie und Phylogenie der Chelicerata. Z. Zool. Syst. Evol-forsch. 17 (85-115), 177-200.). This is a rule of systematics that you have to follow. Now, if you were to find that gill opercula arose convergently, what you could say is that Euchelicerata, as currently diagnosed, is polyphyletic (I already specified that in my first review...). You may also emend the diagnosis to add or replace diagnostic traits. However, and again, you need to be more circumspect here, and admit that your evidence is not good enough to be unambiguous. In MGL107741, those well-defined paddle-shaped elements appressed onto the ventral side could very well be opercula—which is simply defined as a semi-rigid lobate sheet covering lamellae. What you identify as a setose margin is quite tenuous, and would not in fact rule out an operculum in itself.

We agree with the reviewer that considering the state of preservation of these features, we cannot unambiguously confirm they are not opercula. We have accordingly tempered and limited our description and discussion of opisthosomal appendages to their presence and morphology. This is our modified text: “The more posterior opisthosomal appendages in *Dibasterium durgae* and *Offacolus kingi* are modified into gill opercula, but this condition cannot be clearly stated for *Setapedites abundantis* (Fig. 3 and 6, Supplementary Fig. 1G and H), which exhibits a similar proximal portion of the opisthosomal appendages to that of *Habelia optata* (Fig. 3 and 6).”

Nevertheless, and again I am repeating what I wrote in my first review, even if you coded with certainty the absence of operculum in your species, based on parsimony this optimizes as homoplasy! I tried to show that using your Lamsdell dataset and tree result... only to find out that the operculum is not even present in this matrix. How could you test anything related to the evolution of the operculum if you don't even code it in your dataset? So we used the other dataset, from Aria and Caron 2019, and looked at the optimization of opercula for both parsimony and likelihood on the Bayesian result. See attached figure. Likelihood optimization is extremely strict and does not achieve more than ambiguity, but a parsimony-based optimization clearly places the presence of opercula at the Euchelicerata node as retrieved previously in Aria and Caron 2017, 2019, with the ambiguity in Setapedites being a reversal.

Nevertheless, and again I am repeating what I wrote in my first review, even if you coded with certainty the absence of operculum in your species, based on parsimony this optimizes as homoplasy!

As stated above, we have now reduced any statement regarding this character to the maximum that can unambiguously be said about it.

Regarding the phylogenetics matrices used, we indicate (and justify) clearly in our work that we used a dual phylogenetic approach:

- Firstly we assessed the position of our fossil within Arthropoda using Aria and Caron (2019)'s matrix, which as indicated here by the reviewer is well-adapted to look at the optimization of the opercula**
- Secondly, we used Lamsdell (2013)'s matrix to resolve relationships within Chelicerata. And logically, as noticed by the reviewer, this matrix does not contain any character about the opercula as it will be of poor phylogenetic information in this matrix. We never intended to use this matrix to constrain the evolution of the opercula, which we had already done by using the Aria & Caron (2019) matrix.**

That said, we warmly thank the reviewer for running analyses on the optimization of the opercula, and confirming the presence of opercula at the Euchelicerata node.

Similar story for the peculiar exopod attachment condition. This most parsimoniously optimizes as ancestral for panchelicerates—sea spiders notwithstanding—despite you recoding uncertainty in Mollisonia. This evolutionary pattern was put forward as the illustration of the possible mechanism for the loss of the outer branch in the euchelicerate prosoma in Aria and Caron 2017, 2019, an idea that you misleadingly claim to be new here.

This has actually been first recognized by Briggs et al (2012), well before Aria and Caron (2017, 2019). We did not claim it to be a new idea, but just stated that our findings support this condition. To make this clear, we added the citations to the end of every sentence discussing this topic.

Additionally, if Mollisonia's trunk appendages are not multi-lamellate structures akin to book gills, what are they? I am defender of a good science based on open refutation, but refutation should be explicit, fact-based.

It is true that due to the controversial opinions regarding the chelicerate affinities of *Mollisonia* we do not mention its reported book gills in our discussion, but nowhere in

the text we express any scepticism regarding the interpretation of book gills in *Mollisonia*, which we consider to be out of the scope of this work.

> There are issues with the use of “stem” and “crown” terminology throughout, which I think has more to do in fact with the taxonomic issue of what Euchelicerata means as explained above. First, in your Fig. 6, Synziphosurines are not part of the crown, since they left no extant descendants. Second, as I demonstrate above, likelihood and parsimony optimization still places Euchelicerata at the node joining up offacolids and prosomapods.

We agree. In our modified Fig.6 synziphosurines are now clearly indicated as stem euchelicerates and not crown euchelicerates, and Euchelicerata is clearly placed at joining up offacolids and prosomapods.

Third, considering that you’re leaving Chelicerata out of the picture, you need to include early panchelicerates *Mollisonia* and *Habelia* into the stem of euchelicerates.

We agree. We modified Fig. 6 to differentiate lower and upper stem euchelicerates, and removed the non-Euchelicerate from the legend of supplementary Fig. 5.

In summary:

- You need to be more explicit about your rejection of the evidence in *Mollisonia*.
- The evidence is not clear enough to ascertain the absence of opercula. This should be left more open.
- If you actually optimize your character on your tree (which I suggested you to do in the first round but you did not), you find that opercula are present in the common ancestor of Euchelicerata sensus Wygoldt and Paulus and as retrieved by Aria and Caron 2019. This contradicts your statements.
- Same for the prosomal exopod condition: Most parsimoniously this is present in the common ancestor of panchelicerates, again as stated in Aria and Caron 2019.
- Improve your interpretative drawings and reconsider segmental anatomy, as well as prosomal gnathobasipod and endopod identification.

All these points have been addressed above.

I refrained from examining the MS in finer details – language, references, etc. The authors must first address these remaining salient problems, some of which being carried over from the first round of reviews. Should they do so, I expect that only one additional round will be necessary for said details. I want to make clear that I am not trying to impose my views, and I would also rather not spend a week on a review, but I am trying my best to show the authors that the shortcomings I point out are not simply a matter of opinion—as they should not be, on either side.

As stated in our cover letter, we believe that we have substantially revised our manuscript according to most of the reviewer’s new comments, notably by giving up on our interpretation of the opisthosomal appendages. However, unlike what the reviewer is claiming here, his suggested changes in several places of this review amount to forcing us to accept his own views. We cannot compromise our own scientific views of this highly debated topic by agreeing to every change he has suggested, but we have tried to find a middle ground and temper our interpretations by serious and careful consideration of his comments. We thank Cedric for investing so much of his time into these in-depth reviews of our work.

Other specific comments:

I.38: Offacolidae

Amended.

I.51: The statement that it is “obscure” washes away the work on both the Herefordshire and the Burgess Shale. It remains poorly documented, or something equivalent.

Amended.

I.51: Regarding the definition of Euchelicerata here and hereafter, see my comment and correct.

Amended.

I.56: You refer to phylogenetic results that are 10 years old—and in the case of Haug et al. 2012, this is not even a cladistic study. Why not acknowledge that these hypotheses have not been recovered again? What recent phylogenetic study has disputed that Habeliida is the closest sister group to Euchelicerata? (Mollisonia notwithstanding). Even if you reject the evidence provided by Mollisonia, Panchelicerata as defined by Aria and Caron (2019) remains valid, yet there is no mention of it here. This grouping was recovered in most recent cladistic work by Zeng et al., Moysiuk and Caron, etc.

In this section of the introduction we give an overview of the current hypotheses (and divergent opinions) on the origin of chelicerates, and as such we find it totally appropriate to mention views arising from different teams and lines of evidence (even if they are 10 years old). Moreover, the latter mentioned studies by Zeng et al. and Moysiuk & Caron all based their analyses on the same matrix (Aria and Caron 2019) to assess the position of stem arthropods with no impact on Chelicerata, and as such have recovered similar groupings, but cannot be considered as independent confirming lines of evidence.

We agree with the use of Panchelicerata and we have now used this term in the caption to Figure 6 and text we added in the presentation of phylogenetic results that reads: “We then performed a second analysis focusing on Panchelicerata (=Chelicerata+Habeliida), using the matrix from Lamsdell (2013) that includes numerous synziphosurines and euchelicerates, and to which we add *Habelia* and *Mollisonia*.”

I.82: The anal pouch is completely speculative. Without redundancy of evidence, I don't see how this can be differentiated from an artefact. Consider it putative if you want, but just in the figure caption; no need to dwell on that.

In the caption for Fig. 2 (I649) we previously indicated “and the anal pouch in purple”, which we have now modified into “and the possible anal pouch or post-ventral structure (pvs) in purple” to make it clearer that the interpretation of this feature is not unambiguous.

All mentions of this feature in the text are very tempered:

- I82-83, “and possibly an anal pouch”

- I226–229, “A small symmetrical rounded process (pretelsonic process) is often preserved under tergite XI, in some cases appearing longitudinally bisected and ovoid

in shape (ptp in Fig. 2A and 2B, Fig. 229 3A and 3B, Supplementary Fig. 1A–F, Supplementary Fig. 3E and 3F”).

And as such we already meet the request of the reviewer.

Considering that, according to Aria and Caron (2017), *Habelia* possesses an anal pouch of similar morphology, we believe that it is worth keeping our (short) discussion about this feature in Supplementary information.

I.83: “Setapedites abundantis resolves the position of several problematic stem-euchelicerate taxa”: What do you mean? Setapedites has no phylogenetic impact on the placement of early panchelicerates or other offacolids based on recent phylogenetic results. Sys. Pal.: See main comments.

We agree this was unclear. We have changed the sentence to: “*Setapedites abundantis* documents how euchelicerate uniramous prosomal appendages were derived from the appendages of a habeliid ancestor¹² and illuminates the evolution of early tagmosis in euchelicerates.”

I.285: This makes no sense. It’s not because frogs have anterior members homologous with birds that it would play a role in supporting their bird affinity! Also, the statement above is contradictory with what you say elsewhere about gill opercula not being apomorphic of euchelicerates.

We agree that this was not clear. This sentence has been entirely removed. Based on other comments from this reviewer, we no longer discuss the gills in the manuscript.

I.290: Well, no, they are shared with Sanctacaris, and the enlarged gnathobasipods also shared with Mollisonia. Or do you also contest the presence of large gnathobasipods in Mollisonia?

We do not contest the large gnathobasipods in *Mollisonia*. We rephrased our sentence as follows to make it clearer: from “possesses its own derived character not shared with Offacolidae” to “possesses characters not shared with Offacolidae”.

I.301: Not sure what kind of bilobation you mean in *Habelia*.

We refer to the fact that is made of two elements “paddle connected to an attachment podomere”. We have made this clearer by modifying the sentence as follows: “appendages of somite VII made of bipartite paddle-like exopods”.

I.327: No, it does not; quite the opposite in fact. If you posit the clear attachment to a basipod in Setapedites, which I contest, it is another autapomorphy, and a reversal, (like the putative absence of opercula) amid the condition shared by offacolids, habeliids and mollisoniids.

Aria and Caron (2019) indicated that the “frontal appendages in *Sanctacaris* and *Habelia* are in fact quite distinct morphologically from the exopods of *Offacolus* and *Dibasterium*, with long and slender podomeres, giving them indeed a more antennular aspect. Similar to at least *Dibasterium* (but also likely *Offacolus*), however, these rami in *Habelia* are preserved separately from the endopod bundle in non dorso-ventrally-

preserved specimens, suggesting that they do not attach to the limb basis like regular exopods”.

This suggestion has, however, not been unequivocally supported by any fossil evidence. In turn, our fossil shows real evidence for a common basis (if not a true basipod as contested by the reviewer), while this has only been suggested for *Habelia*. Nevertheless, we slightly modified our sentence as follows to temper the identification of a true basipod: “The evidence for the attachment of prosomal biramous limbs onto a common basipodite (or at least a common basis) in *Setapedites abundantis* [...]”.

I.337: The seventh cephalic pair of early panchelicerates has already been argued to be ancestral to these features in Aria and Caron 2017, 2019. This was one of the main findings in these papers...

Acknowledged. Citations added.

Phylogenetic figures: Are you sure you are displaying Maximum clade credibility consensus? The default output from MrBayes is extended majority rule, and the topology coming out of my basic run is the same as yours.

Amended. Thanks for pointing this out. We changed the text to indicate we are using majority rule.

Additionally, happening to be, I think, the “someone like David Legg, Greg Edgecombe, Jason Dunlop, James Lamsdell or such”, I was asked to check whether the authors had fully addressed the comments of reviewer #2, Jakob Vinther. I can’t say about the bottle of “Swizz” white wine, but it seems the authors took care of providing a better rendition of the raw CT data, as well as adding another specimen—which is, I think, what Dr. Vinther suggested. Clarification was also provided regarding the material used, and labelling in Figure 2 was corrected.

We thank the reviewer for taking the time to confirm that our revised manuscript satisfied reviewer #2 (Jakob Vinther)’s comments and suggestions.

Reviewers' Comments:

Reviewer #1:

Remarks to the Author:

I am glad to have received this revised manuscript and that the authors have further considered my comments. While the authors have made interesting points regarding their morpho-anatomical interpretations, and several sections of this work are really excellent, I identified a few strong remaining issues. The problems (one in particular) with the phylogenetic optimization of characters are non-interpretative, and should really be resolved before publication.

I therefore recommend "minor" revisions but these are minor only in number; some of them are still major scientifically speaking. I am attaching an annotated Word file for a number of comments made directly on the manuscript.

I will otherwise not engage in peripheral discussions about what the "majority of the community" supposedly represents or thinks, or debates related to my own work. Note however that the chelicerate affinity of *Mollisonia* was accepted by experienced figures in our field such as Edgecombe (2020 Ann Rev) and that no colleague has so far proposed to reexamine the published material.

Anatomy:

- Obviously, the endopod / exopod interpretation is a challenging issue. The authors have made their points with solid arguments and this can serve as ground for further discussions. Nevertheless, 1) I think it would be nice to report some of what you wrote in your response to the main text, as this can be of interest for other colleagues, and 2) you should explain the inconsistency in length between what you label as exopods in 2H and then in 3B and 4I, J. The short and stout rami you interpret as exopods in MGL.102634 are somehow jutting out extensively beyond the cephalic margin in some other specimens...
- Regarding the body segment counts, it is fine that you disagree with my reinterpretations, but you are still leaving these areas unexplained. For the element anterior to the telson especially, please explain why you think this is part of the telson yet remains physically separated from it in three of your main specimens.

Phylogeny and evolutionary implications:

There is still no optimization of characters on any of the trees. I suppose the diagrams in Fig. 6 palliate this somewhat, but, as shown hereafter, this continues to hamper a correct assessment (for authors and readers) of the evolution of characters.

- About the seventh prosomal appendage... Your paragraph is extremely confusing and doesn't hold phylogenetically, contrary to what you claim. First of all, both *Dibasterium* and *Offacolus* have been described with a seventh prosomal pair of appendages, yet your diagram in Fig. 6 represents *Dibasterium*'s 7th pair as part of the opisthosoma. Although in Briggs et al. 2012 PNAS the ventral-dorsal alignment was discussed, this should be put in the broader perspective of the chilaria and the 7th pair of *Offacolus*, alongside the synziphosurine microtergite. You also seem to be forgetting about *Weinbergina*. In all your topologies, the plesiomorphic 7th prosomal pair continues to be both the most parsimonious and likely scenario. Like I've suggested during 2 rounds of review, just open Mesquite if you don't want to go through the trouble of reconstructing ancestral states with R...
- Illustrating the origin of uniramy in euchelicerates... *Setapedites* fuels discussions about uniramy in other offacolids as well as the question of prosomal limb attachment. In itself, however, I don't see how it illustrates the origin of uniramous appendages in all euchelicerates, as stated in your MS.

You should also comb through the bibliography as I've seen a number of issues (e.g., date missing, extra characters, R Team entry...)

Cédric Aria

Reviewer #4:

Remarks to the Author:

The manuscript describes a new genus and species of synziphosurine chelicerate, named *Setapedites abundantis*, from the Lower Ordovician Fezouata Formation of Morocco and explores its phylogenetic relationships and implications for the early evolution of chelicerates. The manuscript comprises two main components; the description of the fossil material, and the phylogenetic analyses.

The available fossil material supports their interpretations, which are in line with my own thoughts about the material. Particularly of interest are the structure of the prosomal appendages - the elongated chelicera and biramous appendages II-V are particularly important, as this is the first time they have been unequivocally documented in a taxon outside of the Herefordshire Lagerstätte. I do think it would be worthwhile drawing particular attention to the form of the chelicerae given some of the previous suggestions regarding the composition of the appendages (see the list of suggestions below for more detail).

The phylogenies are well-executed, although I do have some concerns about the Lamsdell 2013 matrix - specifically, by rooting the analysis on *Yohoia* the authors essentially constrain the clade comprising *Emeraldella*, *Sidneyia*, and *Olenoides* to be the sister group of chelicerates and force megacheirans to be paraphyletic. These are obviously all hypotheses that have been suggested at one time or another, although I do think that a trilobitomorph affinity for chelicerates is not particularly well supported in the current literature and it may be best to remove those three taxa from the analysis as they may be exerting an undue influence on character polarity. The authors should also remove *Willwerathia* from the analysis, as the taxon was recently excluded from Chelicerata (see Lamsdell 2020; <https://doi.org/10.1007/s12542-019-00493-8>).

Finally, in the comparison of *Setapedites* to other taxa, the authors may want to note that the telson of *Bunodes* lacks a bifurcate termination and is quite unlike that of offacolids. The authors should probably also discuss *Satrocercus*, which Legg (2014; 10.1007/s00114-014-1245-4) retrieved as the sister taxon to *Offacolus* and Jago et al. (2016; 10.1111/pala.12243) considered to be a chelicerate. While I do not find the chelicerate affinities of *Satrocercus* particularly compelling, the fact that it has implicitly been considered an offacolid in the past means it should be discussed at least in passing.

Overall this is a good manuscript, and I am glad to see the Fezouata synziphosurine finally get a formal description. The figures are clear and the photographs excellent given the very small size of the material. I just have a handful of suggestions that I hope the authors will take into account, which can be summarized as:

1. Make specific note of the fact that the elongated chelicera are composed of a few elongated articles and are not in fact antenna-like, which has been suggested by previous authors (e.g. Legg 2014).
2. Remove *Willwerathia*, *Olenoides*, *Emeraldella*, and *Sidneyia* from the phylogenetic analysis building upon the Lamsdell 2013 matrix.
3. Note the form of the telson of *Bunodes*.
4. Discuss the affinities of *Satrocercus* given its previously suggested relationship to *Offacolus*.

Point-by-point responses to the Referees' comments and suggestions.

We present the referees' comments in normal typeface and our responses in blue.

Reviewer #1 (Remarks to the Author):

I am glad to have received this revised manuscript and that the authors have further considered my comments. While the authors have made interesting points regarding their morpho-anatomical interpretations, and several sections of this work are really excellent, I identified a few strong remaining issues. The problems (one in particular) with the phylogenetic optimization of characters are non-interpretative, and should really be resolved before publication.

I therefore recommend "minor" revisions but these are minor only in number; some of them are still major scientifically speaking. I am attaching an annotated Word file for a number of comments made directly on the manuscript.

I will otherwise not engage in peripheral discussions about what the "majority of the community" supposedly represents or thinks, or debates related to my own work. Note however that the chelicerate affinity of *Mollisonia* was accepted by experienced figures in our field such as Edgecombe (2020 Ann Rev) and that no colleague has so far proposed to reexamine the published material.

We are happy that our edited version represents an improvement to our reviewer. We have now incorporated a phylogenetic optimization of characters and addressed all the other major and minor suggestions received in this round of review, with the sole exception of the comment "About the seventh prosomal appendage..." which we were able to amend only partially, but for which we tried to present below the most complete explanation of our thoughts. Finally, we apologize for using the argument of "majority of the community". We are aware that is not a scientific argumentation of our interpretation, but we mean it as an argument of general support.

Anatomy:

- Obviously, the endopod / exopod interpretation is a challenging issue. The authors have made their points with solid arguments and this can serve as ground for further discussions. Nevertheless, 1) I think it would be nice to report some of what you wrote in your response to the main text, as this can be of interest for other colleagues, and 2) you should explain the inconsistency in length between what you label as exopods in 2H and then in 3B and 4I, J. The short and stout rami you interpret as exopods in MGL.102634 are somehow jutting out extensively beyond the cephalic margin in some other specimens...

This is an important point to address, so we added the following to explain both the points raised by the reviewer: "This is especially true for *Setapedites* and *Offacolus kingi* exopods 2 to 5 considering the exopods shown in Figure 2G-H. The mesial position of the exopods could cast some doubts on their exopodial nature, however both endopods and exopods have their proximal-most parts very close to each other (visible in Fig. 4D-J) and this has been supposed to be the case in other offacolids too¹⁵⁻¹⁷. The length of the most well-preserved exopod in this specimen (see also Supplementary Fig. 3) also appears shorter than the exopods preserved in other specimens and extending outside the prosomal shield (e.g. Fig. 1A and B), which is likely owing to its preservation in a bent position with an angle parallel to the matrix and the probability of it being the fourth or even the fifth exopod."

- Regarding the body segment counts, it is fine that you disagree with my reinterpretations, but you are still leaving these areas unexplained. For the element anterior to the telson especially, please explain why you think this is part of the telson yet remains physically separated from it in three of your main specimens.

We have now adjusted the figures and the text to better clarify this topic. Figure 1 now has labels on each of the three specimens to show the authors' interpretation of the segments, which interpret the head of the telson being separated only in the specimen shown in Figure 1A. We also inserted into the description in the main text the following: "It is internally articulated with the last abdomen segment by an enlarged head (Fig. 1A and B)."

Phylogeny and evolutionary implications:

There is still no optimization of characters on any of the trees. I suppose the diagrams in Fig. 6 palliate this somewhat, but, as shown hereafter, this continues to hamper a correct assessment (for authors and readers) of the evolution of characters.

We have added an optimization of characters for the opercula (Supplementary Figure 18), to better show that opercula are supposed to be present in *Setapedites* on the basis of the reconstructed ancestral state analysis as suggested by the reviewer. We hope it is now clear for the reviewer and the readers that on the basis of our phylogenetic analysis, this character is expected to be there in *Setapedites*. We cannot code it as present because we have not observed it in our fossils, so the coding for the opercula in *Setapedites* stays ?, since it is not observed. We hope this clarifies the point so the readers can understand it is supposed to be there according to the phylogenetic analyses, especially since we also removed mention of the possibility of this character being truly absent from the anatomy of *Setapedites* in earlier rounds of review. Regarding the reconstructed ancestral state for the prosomal tagmosis, we have performed it and included it as supplementary material as well (Supplementary Figure 16.17), however, see the comment below.

- About the seventh prosomal appendage... Your paragraph is extremely confusing and doesn't hold phylogenetically, contrary to what you claim. First of all, both *Dibasterium* and *Offacolus* have been described with a seventh prosomal pair of appendages, yet your diagram in Fig. 6 represents *Dibasterium*'s 7th pair as part of the opisthosoma. Although in Briggs et al. 2012 PNAS the ventral-dorsal alignment was discussed, this should be put in the broader perspective of the chilaria and the 7th pair of *Offacolus*, alongside the synziphosurine microtergite. You also seem to be forgetting about *Weinbergina*. In all your topologies, the plesiomorphic 7th prosomal pair continues to be both the most parsimonious and likely scenario. Like I've suggested during 2 rounds of review, just open Mesquite if you don't want to go through the trouble of reconstructing ancestral states with R...

We thank the reviewer for pointing out that this paragraph was unclear. We have revised this paragraph trying to enhance its clarity and modify the likely state for *Offacolidae* seventh pair of appendages as plesiomorphic. The paragraph about appendages seven now reads as this:

"Regarding the evolution of the seventh pair, and following opisthosomal, appendages, most of our phylogenetic analyses (Fig. 6, Supplementary Figs. 5, 6 and 8) retrieve the wide paddle-like morphology of the exopods of somite VII in *Setapedites abundantis* and *Offacolus kingi* as likely plesiomorphic for the *Offacolidae*, considering the similar morphology of those exopods in *Habelia optata*. Given that *Dibasterium durgae* possesses a reduced condition of the exopod of somite VII 17, it has been considered as homologous to the xiphosurid chilaria 46 and the metastoma of eurypterids and chasmataspidids 27. While we agree on the homology between those somites and related appendages, the peculiar morphology of the seventh pair of appendages in *Dibasterium durgae* is more likely autoapomorphic, if considered into the broader *Offacolidae* array of morphologies for these appendages."

Reconstruction of the ancestral state for the head tagmosis has been performed as requested too (Supplementary Figures 16, 17).

However, a reevaluation of the concept of tagma was in order and we couldn't state that neither *Offacolus* nor *Dibasterium* have a seven somite prosoma.

This is a rather crucial topic and we are grateful to the reviewer for raising it because as stated in Arthropod Segmentation and Tagmosis by Fusco and Minelli p. 212 (in Arthropod Biology and Evolution, edit. Minelli, Boxshall and Fusco 2016): "Although used to encapsulate the main features that characterize body architecture, the tagmata of a given animal often have boundaries whose positions are somehow disputable, **depending on the definition of tagma adopted** and on the interpretation of the body structures present in proximity to the putative boundaries themselves, in the context of the animal life history, morphology, development and evolution. "

In order to avoid future confusion here we follow with **1) what we mean by tagma, prosoma, opisthosoma, prosomal appendages and opisthosomal appendages; 2) recapitulate what we have in *Setapedites* and what is present in *Dibasterium*, *Offacolus*, *Weinbergina* and *Habelia* according with the definition of tagma. We have also double-checked the correct usage of the two words (cephalothorax and prosoma) along the text.**

N.B. Accordingly with Dunlop & Lamsdell 2016, we refer to the ocular somite as somite 0, so to us the seventh somite corresponds to the eighth somite of Aria & Caron 2017, 2019.

1) The meaning of tagma, prosoma, opisthosoma, prosomal appendages and opisthosomal appendages:

Tagma: According to what we found to be the most consistent definition in the literature, (found in the paper Segmentation and tagmosis in Chelicerata, Dunlop and Lamsdell 2016) a tagma is defined as follows:

"Drawing on an older concept developed by van der Hammen (1963) for mites, Lamsdell (2013, p. 4) proposed adopting the term 'pseudotagma' for units defined by differentiation of the tergites or sternites without an associated change in form or function of the appendages. By contrast **true tagmata would be defined as regions of functional specialization**, which in arthropods is predominantly mediated through modification or suppression of the appendages. Applying this concept to Chelicerata we can argue that the prosoma and opisthosoma represent true tagmata because there is a fundamental split (Fig. 2, Fig. 3, Fig. 4, Fig. 5) between the **feeding and locomotory limbs of the prosoma** and the **flattened genital/gill opercula of the opisthosoma** in horseshoe crabs or the largely suppressed or highly-modified appendages of the opisthosoma in arachnids."

In summary, our definition of tagma is the same of Lamsdell 2013 and Dunlop and Lamsdell 2016: "regions of functional specialization, which in arthropods is predominantly mediated through modification or suppression of the appendages" while we consider the definition of pseudotagma the following: "the units defined by differentiation of the tergites or sternites without an associated change in form or function of the appendages".

While tagma and pseudotagma define the body parts using two different criteria, often they may concur to define the same boundary, and if possible they should both be used to define the somites belonging to the prosoma and the somites belonging to the opisthosoma. On top of this, developmental consideration may concur to define the tagma border, has is the case in *Limulus*.

Prosoma and opisthosoma: the two tagmata of chelicerates (see definition of tagma), as they are defined by the presence of **"regions of functional specialization**, which in arthropods is predominantly mediated through modification or suppression of the appendages".

Prosomal appendages and opisthosomal appendages: Following the definition of prosoma and opisthosoma, prosomal appendages will be defined by a “functional specialization, which in arthropods is predominantly mediated through modification of them (the appendages)”. As exemplified from these examples from Dunlop and Lamsdell 2016 “Applying this concept to Chelicerata we can argue that the prosoma and opisthosoma represent true tagmata because there is a fundamental split (Fig. 2, Fig. 3, Fig. 4, Fig. 5) between the **feeding and locomotory limbs of the prosoma** and the **flattened genital/gill opercula of the opisthosoma** in horseshoe crabs”.

2) What we do have in *Setapedites*, *Dibasterium*, *Offacolus*, *Limulus* and *Weinbergina* and *Habelia*

We endorse the above-recapitulated definitions of tagma, prosoma, opisthosoma, prosomal appendages and opisthosomal appendages. As such, it should be clear now that **IF** we follow the definition of Lamsdell 2013, the seventh somite is part of the opisthosoma in *Setapedites*, *Dibasterium* and *Offacolus*. What the reviewer said, “First of all, both *Dibasterium* and *Offacolus* have been described with a seventh prosomal pair of appendages, yet your diagram in Fig. 6 represents *Dibasterium*’s 7th pair as part of the opisthosoma” is true. However, both *Dibasterium* and *Offacolus* were published before Lamsdell 2013 and Dunlop and Lamsdell 2016, the terminology and evolutionary implication of which we are following. In Dunlop and Lamsdell 2016, on which our reconstruction in Figure 6 is based, the seventh somite of *Dibasterium* and *Offacolus* is considered to belong to the opisthosoma on the basis of the associated appendages which, even if showing peculiar anatomical function are rather more similar to the “flattened opercula of the opisthosoma” than to the “feeding and locomotory limbs of the prosoma” (our Figure 6 shows these details and is based on what Dunlop and Lamsdell 2016 reported, without edits in this document as Figure 1). The seventh pair of appendages in *Offacolus* and *Dibasterium* in fact share with the opisthosomal appendages the following characteristics:

Figure 1. Figure 1. Schematic reconstructions in Dunlop and Lamsdell 2016 from which our Figure 6 is based on. In blue prosomal somites and appendages, in yellow opisthosomal somites and appendages.

1) only exopods are expressed, 2) they are flat, 3) they lack the endopods, 4) they do not serve a walking function. All of this is true for *Setapedites* as well, so we attributed an opisthosomal affinity to the seventh somite and associated appendages. In the case of *Setapedites*, even if we consider the alternative definition of tagma (according to Dunlop and Lamsdell 2016 now used as the definition of pseudotagma), the seventh somite clearly belongs to the opisthosoma, bearing fully expressed tergites with pleura like the other opisthosomal tergites. Also in *Dibasterium*, the seventh somite expresses a tergite, even if reduced.

The case of *Limulus* is well described in several papers, and thanks to developmental data we know that the seventh somite appendages develop from opisthosomal-like appendages. In this context, as depicted in Figure 2 of this document, the **pseudotagma cephalothorax encompasses** the first seven somites, and so the chilaria, while the **tagma prosoma** encompasses six somites. This terminology is used as an example also to define the position of the chilaria in Bicknell et al. 2018 (A 3D anatomical atlas of appendage musculature in the chelicerate arthropod *Limulus polyphemus*) in which it is stated that the chilaria belongs to the cephalothorax but is also the first of the opisthosoma. “These small, spine-bearing appendages are the most posterior, heavily modified and unsegmented cephalothoracic appendages that are also considered to be the first of the opisthosomal appendages (Fig 2G and 2H) [40, 47]. The chilaria have no internal musculature and function mostly as a means of preventing food items escaping behind the pushing legs [47].”

Following the stated definition of tagma and associated appendages, *Habelia*'s seventh pair of appendages should be considered part of the opisthosoma as well. However, we decided to follow the interpretation of the reviewer, who in his paper (Aria & Caron 2017) interpreted the seventh somite of *Habelia* as a prosomal somite. In fact, even though the seventh pair of appendages in *Habelia* are clearly of the same nature as the opisthosomal appendages they are interpreted as prosomal appendages in this paper. This association has been made on the basis of the proximity with the prosomal appendages and on the dorsal tergite being part of the head shield. While we consider this to fit with the definition of a **pseudotagma and not of a tagma**, and as such should not be accepted as a reason to include the seventh somite into the prosoma, the functional explanation of how the seventh pair of endopods may be involved in food manipulation, is the perfect example of “**feeding and locomotory limbs of the prosoma**” and so, we accepted *Habelia*'s seventh somite as part of the prosoma for the strong evidence of “**regions of functional specialization**” in the somite from one to seven of *Habelia*. However, according to the definition of tagma we follow, the seven-somite prosoma of *Habelia* is the most debatable tagma here discussed and to consider the seventh somite of *Habelia* as part of the opisthosoma is still a valid option based on the uniformity its appendages have with the opisthosomal appendages. **Contrarily, a seven-tergite cephalothorax in *Habelia* is what we agree on and according to the reconstruction of ancestral state is a condition emerging at the root of euchlicerates.**

Weinbergina has been ignored in this debate for the same reason for which we emended its coding in the matrix used for the phylogenetic analyses. Selden et al 2015 pp. 647 stated:

“Re-evaluation of this species suggests that the structure of the prosomal appendages may be more similar to those of *Dibasterium* and *Offacolus*, each limb being biramous with a potentially chelate endopod and a pediform exopod. Support for this interpretation stems from Fig. 6 of Sturmer & Bergstrom (1981) and Fig. 4 of Moore et al. (2005), which show the endopods curving ventrally on the specimen, while the supposed terminations of these limbs project outward from the carapace; these terminations may, in fact, represent the distal portions of the exopods, while the distal podomeres of the endopods have not been preserved.

Figure 2. Difference between the term cephalothorax and prosoma according to Lamsdell 2013

Fig. 7 of Sturmer & Bergstrom (1981) and Fig. 2 of Moore et al. (2005) also both show specimens with a series of potentially chelate appendages contracted and curving inwards on the prosoma while a second series of appendages more closely resembling the setiferous exopods of *Offacolus* and *Dibasterium* originate dorsally from the insertion point of the inwardly angling appendages and project outwards from the prosoma. It is possible that the first set of appendages represents the endopods, while the second set represents the exopods. Further study of *Weinbergina* will be required to ascertain the exact situation of the prosomal limbs, but it is worth noting that coding *Weinbergina* according to this revised interpretation results in no change to its position within the tree”.

This also leads to the implication reported in Dunlop & Lamsdell 2016 pp. 405 “It should, however, be noted that Selden et al. (2015) suggested that the limbs of *Weinbergina* may in fact be biramous; referring to several published figures in support of this interpretation. Further study of the available specimens of *Weinbergina* is needed to ascertain the veracity of these suggestions, however if they are accurate this would suggest that *Weinbergina* has a closer relationship to *Offacolus* and *Dibasterium* than to Prosomapoda, and would be considered a derivative of the euchelicerate stem lineage. This scheme may also result in the apparent limb of somite VII instead being the uniramous limb of somite VI.”

As widely discussed above, to include a taxon with currently uncertain appendicular anatomy into this debate seemed pointless due to the much better-known anatomy of *Dibasterium* and *Offacolus*.

Finally, with regards to the evolution of the cephalothorax, we do agree with the reviewer on the fact that given our phylogenetic analyses, the most parsimonious scenario for its evolution is at the base of euchelicerates given *Habelia*'s anatomy. This is now well stated in the text (see below) but alongside a discussion of the importance of the fully expressed seventh tergites of *Setapedites* as well, in the broad picture of chelicerates tagmosis and pseudotagmosis evolution.

It is important to note that the authors are aware that the seventh somite itself is different from the other opisthosomal somite in *Offacolidae* as well as in many other euchelicerates. Being the seventh, the somite of transition from one tagma to the other is quite unique in its anatomy respective to both the prosoma and the opisthosoma. As it is reasonable to expect in biology, our systematic definition does not perfectly reflect the reality in all its complexity, and we have to accept that the concept of prosoma and opisthosoma are human constructs that are simplifications of much more complex and much less symmetrical anatomical features resulting from contingent evolutionary processes. As such, we hope that by clearly stating which definition of tagma, prosoma, opisthosoma, prosomal appendages, and opisthosomal appendages we use in this manuscript should end this debate and explain on what basis we consider the seventh somite as belonging, after all, to the opisthosoma in *Setapedites*, *Dibasterium*, *Offacolus* and *Limulus*.

We have added the following to the main manuscript text to clarify the different tagmosis definitions, and now the paragraph reads as follows:

“Tagmosis, the seventh somite and ancestral reconstruction of the euchelicerate ground plan

Setapedites possesses a seventh somite with a distinct tergite and appendages with a morphology similar to the opisthosomal appendages. Within *Offacolidae*, *Dibasterium* has a seventh somite bearing a micro-tergite and appendages with a reduced distinct morphology (unlike either the prosomal or opisthosomal appendages), and in *Offacolus* the seventh somite lacks a distinct tergite and bears appendages similar to those of the opisthosoma (lack of endopods and flat exopods). Likewise, *Habelia* has also been described as possessing a seventh somite lacking a distinct tergite and bearing appendages with morphology similar to the opisthosomal appendages. *Offacolus* and *Habelia* also both have a prosomal dorsal shield covering the first seven pairs of appendages. Despite these similar conditions, Dunlop & Lamsdell (2017) interpret the prosoma of *Offacolus* as having six somites on the basis of the morphology of the appendages, whereas Aria

& Caron 2017 interprets the prosoma of *Habelia* as having seven somites on the basis of the cephalic shield covering and shared specialized feeding function of the first seven appendage pairs despite their different morphology.

Discussions of ancestral reconstructions of the euchelicerate ground plan and the number of somites found in the prosoma and the opisthosoma at the base of Euchelicerata require a clear definition of tagma versus pseudotagma (see Supplementary Information for additional discussion about the definition of tagma versus pseudotagma). In this manuscript we follow Lamsdell (2013) and Dunlop & Lamsdell (2017) in distinguishing between a tagma as a region of functional specialization predominantly identified by modification or suppression of appendages, whereas a pseudotagma shows differentiation in the tergites or sternites without associated change in the form or function of appendages. In this framework, *Setapedites* is important because it is a stem lineage euchelicerate with a seventh somite that has a distinct and well-defined separate tergite and an opisthosomal-like appendage (lack of endopods and flat exopods) associated with it, such that the seventh somite cannot belong to the prosoma. *Setapedites*, and probably *Dibasterium* with its micro-tergite, have a cephalic pseudotagma that matches the cephalic tagma, with both six tergites incorporated into the shield overlapping six pairs of appendages, whereas *Offacolus* has a seven tergites pseudotagma and six somites tagma. Regarding *Habelia*, what Aria & Caron (2017) interpreted as a seven-somite prosoma (tagma) is what we interpret as a seven tergite dorsal shield (pseudotagma), and we regard the seventh pair of appendages, in this basal taxon, as unclear in whether they are associated with the prosoma or opisthosomal. Our ancestral state reconstruction supports a seven tergite dorsal shield at the base of Euchelicerata, but not necessarily a seven-somite prosoma tagma, as stated by Aria & Caron (2017, 2019) (see Supplementary Fig. 16 for the character history reconstruction of the cephalic pseudotagma and 17 for the character history reconstruction of the cephalic tagma). What we see for these stem lineage euchelicerate taxa is a myriad morphologies for the seventh somite and associated appendages, underlining the morphological plasticity of this segment at the boundary between two major body tagma."

To further elucidate the tagmosis definition, also the following paragraph has been added de-novo into the supplemental discussions: "**Definitions of tagmosis and related terms**

Debates about the nature of the ancestral euchelicerate segmentation in the head and body hinge partly on the terminology used, and variable definitions for key terms such as tagmosis, tagma, pseudotagma, prosoma, opisthosoma, prosomal appendages and opisthosomal appendages. As stated in by Fusco and Minelli (2016, p. 212) in ²⁹, "Although used to encapsulate the main features that characterize body architecture, the tagmata of a given animal often have boundaries whose positions are somehow disputable, depending on the definition of tagma adopted and on the interpretation of the body structures present in proximity to the putative boundaries themselves, in the context of the animal life history, morphology, development and evolution". Reviews of the definition of tagma in euchelicerates is available in Lamsdell (2013, p. 4) ²⁸ which proposed to adopt the "term 'pseudotagmata' for units defined by differentiation of the tergites or sternites without an associated change in form or function of the appendages. By contrast true tagmata would be defined as regions of functional specialization, which in arthropods is predominantly mediated through modification or suppression of the appendages.". Conversely, pseudotagma (and so a cephalothorax) has been reported as "the units defined by differentiation of the tergites or sternites without an associated change in form or function of the appendages". This same definition has been adopted in a review of the tagmosis in chelicerates by Dunlop and Lamsdel (2017) ³⁰. The definition of tagma used in the phylogenies of Aria & Caron (2017, 2019) ^{5,27} is a mixture of the definition of tagma and pseudotagma, described as "somital head (as tagma I) defined by series of appendages and/or external segmentation" (Aria & Caron 2017 page 13 of supplemental

information⁵; Aria & Caron 2019 Page 11 of supplemental information²⁷). In this paper, we follow the definition found in Dunlop & Lamsdell 2017³⁰. While this creates no controversies in the coding of characters in the matrix from Lamsdell 2013²⁸, it may create some misunderstandings for our coding on the matrix from Aria & Caron (2019)²⁷. For this last matrix we decided to do not update the coding for the character accounting for the tagmosis for two main reasons. Firstly, tagma and pseudotagma are just two different characters with no intrinsic value of being more or less informative, as far as is clear what we are meaning by each of them. Secondly for the anatomy of *Habelia optata*, we do recognize that not only the pseudotagmatization of this species may incorporate seven tergites, but also that the seventh somite appendages may be, as described in Aria & Caron (2017)⁵ be functionally involved in the head tagma function. As such not having examine the material ourselves we keep the exact coding of Aria & Caron (2019)²⁷ for this character in the matrix we used. Lastly, it should be noted that currently the anatomy of *Weinbergina*, used to sustain a seven-somite head tagma at the root of euchelicerates is considered by many as in need for a review^{30,31}.

- Illustrating the origin of uniramy in euchelicerates... Setapedites fuels discussions about uniramy in other offacolids as well as the question of prosomal limb attachment. In itself, however, I don't see how it illustrates the origin of uniramous appendages in all euchelicerates, as stated in your MS.

Amended. We refreshed this sentence to say "while also **contributing** to our understanding of the origin of euchelicerate uniramous prosomal appendages".

You should also comb through the bibliography as I've seen a number of issues (e.g., date missing, extra characters, R Team entry...)

Amended.

Cédric Aria

Reviewer #4 (Remarks to the Author):

The manuscript describes a new genus and species of synziphosurine chelicerate, named *Setapedites abundantis*, from the Lower Ordovician Fezouata Formation of Morocco and explores its phylogenetic relationships and implications for the early evolution of chelicerates. The manuscript comprises two main components; the description of the fossil material, and the phylogenetic analyses.

The available fossil material supports their interpretations, which are in line with my own thoughts about the material. Particularly of interest are the structure of the prosomal appendages - the elongated chelicera and biramous appendages II-V are particularly important, as this is the first time they have been unequivocally documented in a taxon outside of the Herefordshire Lagerstätte. I do think it would be worthwhile drawing particular attention to the form of the chelicerae given some of the previous

suggestions regarding the composition of the appendages (see the list of suggestions below for more detail).

The phylogenies are well-executed, although I do have some concerns about the Lamsdell 2013 matrix - specifically, by rooting the analysis on *Yohoia* the authors essentially constrain the clade comprising *Emeraldella*, *Sidneyia*, and *Olenoides* to be the sister group of chelicerates and force megacheirans to be paraphyletic. These are obviously all hypotheses that have been suggested at one time or another, although I do think that a trilobitormorph affinity for chelicerates is not particularly well supported in the current literature and it may be best to remove those three taxa from the analysis as they may be exerting an undue influence on character polarity. The authors should also remove *Willwerathia* from the analysis, as the taxon was recently excluded from Chelicerata (see Lamsdell 2020; <https://doi.org/10.1007/s12542-019-00493-8>).

Finally, in the comparison of *Setapedites* to other taxa, the authors may want to note that the telson of *Bunodes* lacks a bifurcate termination and is quite unlike that of offacolids. The authors should probably also discuss *Satrocercus*, which Legg (2014; 10.1007/s00114-014-1245-4) retrieved as the sister taxon to *Offacolus* and Jago et al. (2016; 10.1111/pala.12243) considered to be a chelicerate. While I do not find the chelicerate affinities of *Satrocercus* particularly compelling, the fact that it has implicitly been considered an offacolid in the past means it should be discussed at least in passing.

Overall this is a good manuscript, and I am glad to see the *Fezouata* synziphosurine finally get a formal description. The figures are clear and the photographs excellent given the very small size of the material. I just have a handful of suggestions that I hope the authors will take into account, which can be summarized as:

We are happy that our new reviewer evaluated our manuscript positively. Accordingly, with the suggestion received, we have now incorporated a specific note of the fact that the elongated chelicera are composed of a few elongated articles in *Setapedites*, we have provided new phylogenetic analyses and modified old on according with suggestions, discussed the telson of *Bunodes* and finally discussed *Satrocercus*.

1. Make specific note of the fact that the elongated chelicera are composed of a few elongated articles and are not in fact antenna-like, which has been suggested by previous authors (e.g. Legg 2014).

We have now included a statement on the anatomy of the chelicera: "The first pair of uniramous appendages in *Setapedites abundantis*, although rarely preserved, appear to be composed of few elongated articles, confirming their elongated chelicera interpretation in other offacolids and discharging the antenna-like hypothesis (Fig. 2C-D)."

2. Remove *Willwerathia*, *Olenoides*, *Emeraldella*, and *Sidneyia* from the phylogenetic analysis building upon the Lamsdell 2013 matrix.

Willwerathia has been removed from all phylogenies. Regarding the outgroups, we agree that using *Yohoia* as an outgroup creates this problem and we were advised in a previous round of review to remove *Fuxianhuia protensa* as outgroup. Our original intent was to leave the outgroups unchanged. We have included a new set of phylogenetic analyses in the supplementary material to show our result of a phylogenetic analysis without the Artiopoda as suggested by the reviewer. However, we did not use this as

the main tree in our paper. This is because to remove from our matrix 5 out of 40 taxa, of which 4 are outgroups will have a negative effect on the stability of the analyses (SFig.14). So even though we agree with the reviewer, we find that leaving only the megacheirans as an outgroup of the chelicerates is not enough to polarize the analyses correctly, and we proposed a tree without artiopodans in the supplementary material (SFig. 12-15) to show that this has no major effects on our conclusions.

3. Note the form of the telson of *Bunodes*.

We have now included in the supplementary discussion on the other Synziphosurines this statement to underline the diverse telson morphology of offacolids and *Bunodes*: "However, it is differentiated by the presence of a reduced first tergite (microtergite) and the total number of tergites and a thicker, more conical telson devoid of bifurcation."

4. Discuss the affinities of *Sarotrocercus* given its previously suggested relationship to *Offacolus*.

We have also implemented in the same section of the supplementary discussions a comparison with *Sarotrocercus*: " Finally, we consider *Sarotrocercus oblitus*, which has been recovered in some phylogenies as possible a chelicerate^{8,25}. According to its most recent redescription²⁶, even though this taxon resembles the Offacolidae anatomy dorsally, especially *Offacolus kingi*, its ventral anatomy differs from this group in many aspects such as: a lower head tagmatization, reduced exopods, and absence of a division of the trunk into pre-abdomen and abdomen. However, the trunk exopods of *Sarotrocercus oblitus* may represent an early evolutionary step towards the opercula with the possible loss or strong reduction of the endopods. Confirmation of its chelicerates affinities is subject to further study."

Reviewers' Comments:

Reviewer #4:

Remarks to the Author:

The manuscript is an important contribution to our understanding of early chelicerate evolution. The revisions address all of the comments from my previous review, and I have no further suggested revisions or additions.